# Concept-Guided Interpretability via Neural Chunking

**Shuchen Wu[1,2], Stephan Alaniz[3], Shyamgopal Karthik[4],**
**Peter Dayan[5], Eric Schulz[6], Zeynep Akata[4]**
[1]Allen Institute, [2]University of Washington, [3]Télécom Paris, Institut Polytechnique de Paris,
[4] Institute of Explainable Machine Learning, Helmholtz Munich
[5] Department of Computational Neuroscience,
Max Planck Institute for Biological Cybernetics
[6] Institute for Human-Centered AI, Helmholtz Munich
`shuchen.wu@alleninstitute.org`

## Abstract

Neural networks are often described as black boxes, reflecting the significant challenge of understanding their internal workings and interactions. We propose a different perspective that challenges the prevailing view: rather than being inscrutable, neural networks exhibit patterns in their raw population activity that mirror regularities in the training data. We refer to this as the *Reflection Hypothesis* and provide evidence for this phenomenon in both simple recurrent neural networks (RNNs) and complex large language models (LLMs). Building on this insight, we propose to leverage our cognitive tendency of *chunking* to segment high-dimensional neural population dynamics into interpretable units that reflect underlying concepts. We propose three methods to extract recurring chunks on a neural population level, complementing each other based on label availability and neural data dimensionality. Discrete sequence chunking (DSC) learns a dictionary of entities in a lower-dimensional neural space; population averaging (PA) extracts recurring entities that correspond to known labels; and unsupervised chunk discovery (UCD) can be used when labels are absent. We demonstrate the effectiveness of these methods in extracting concept-encoding entities agnostic to model architectures. These concepts can be both concrete (words), abstract (POS tags), or structural (narrative schema). Additionally, we show that extracted chunks play a causal role in network behavior, as grafting them leads to controlled and predictable changes in the model's behavior. Our work points to a new direction for interpretability, one that harnesses both cognitive principles and the structure of naturalistic data to reveal the hidden computations of complex learning systems, gradually transforming them from black boxes into systems we can begin to understand. Implementation and code are publicly available at `https://github.com/swu32/Chunk-Interpretability`.

## 1 Introduction

Neural networks are known as "black boxes" [80, 71, 73, 93, 56, 25], highlighting the inherent challenge to make sense of a vast number of computing units. Their computations are performed by up to billions of interacting components, in stark contrast to the traditional scientific practice of modeling phenomena through a number of well-defined symbolic entities in physics or mathematics.

A substantial portion of the interpretability challenge is cognitive [1, 60] and demands asking the question: what makes high-dimensional data meaningful for cognition? And what constitutes the basic units and entities for perception that underlie the constitution of the symbols that we are adept at studying in physics or math?

39th Conference on Neural Information Processing Systems (NeurIPS 2025).

We present a novel perspective on interpreting artificial neural networks by referencing how *cognitive entities* arise from sensory data. Similar to the vast neural activities, perceptual data that flood into our sensory stream is similarly high-dimensional and complex. However, there is an inherent tendency for cognition to rapidly discern recurring patterns in perceptual sequences as chunks and entities ([91, 90, 4, 13]) - a process that even infants do with minimal exposure [75, 76, 3]. Our perceptual and cognitive system adeptly organizes the the overwhelming flow of the sensory stream into a structured representation of entities, relations, and events over time [37, 35, 23, 26, 46, 14, 89, 88].

Psychologists have long postulated that identifying patterns in perception serves as an inverse model to uncover structured reality, in which recurring elements often reflect shared causal mechanisms [64, 7]. We termed this the "reflection hypothesis" and examined if the neural network's hidden activity reflects the regularities in the data it learns to predict, first in RNNs trained to predict sequences with known regularities, and then in large language models. Evidence that structured information exists in high-dimensional neural data suggests the possibility of leveraging our cognitive tendency to chunk information and discern recurring patterns and entities directly from neural population activities. Hence, we developed several methods to extract these recurring entities from low to high dimensions of neural population activities, complementing each other in their usage cases. We show the advantage of this simple method over popular interpretability approaches. Our findings suggest a novel interpretability framework that leverages the cognitive principle of chunking to find emerging entities within artificial neural activities, paving the way for future work that decomposes network computation into interactions and relations between symbolic entities.

## 2   Related Work

Most interpretability approaches hold a salient agreement on *what is interpretable* and *what to interpret*. *What is interpretable* is influenced by models in physics and mathematics, where operations and derivations are framed around the manipulation of a small set of well-defined symbols. Hence, interpretable concepts are confined to word-level or token-level description, and methods try to learn a mapping between the neural activities and the target interpretable concept [33, 94, 9, 10, 69, 20].

Current methods on *what to interpret* to understand the computations inside a neural network are heavily influenced by neuroscience: either on the level of neurons as a computation unit or in low-dimensional neural activity descriptions. The earliest interpretability approaches, inspired by discoveries of "grandmother cells" and "Jennifer Aniston neurons" inside the brain, focused on understanding the semantic meanings that drive the activity of individual neurons. Similarly, studies in ANNs, from BERT to GPT, have identified specific neurons and attention heads whose activations correlate with semantic meanings in the data [68, 25, 85, 57, 8, 36, 65, 63, 72, 43]. Sparse autoencoders (SAEs)can be seen as an intermediate step that encourage the hidden neurons to be more monosemantic [32, 54, 18, 15, 44]. Thereby, an autoencoder maps neural activities of a hidden unit layer to a much larger number of intermediate hidden units while encouraging a small number of them to be active. This way, the activity of the target hidden layer can be represented as a superposition of several individual neurons within the SAE. Other approaches such as representation engineering captures the distinct neural activity corresponding to the target concept or function, such as bias or truthfulness [94]. Then, it uses a linear model to identify the neural activity direction that predicts the concept under question or interferes with the network behavior.

The current interpretability approaches that study language-based descriptions as conceptual entities in terms of individual neurons or low-dimensional projections suffer from limitations on both ends: meanings are finite, and individual neurons are limited in their expressiveness and may not map crisply to these predefined conceptual meanings. Just like the failure in physics to have a closed-form description of motion beyond two interacting bodies [79], confined, symbolic definitions of interpretation have inherent limitations in precision. This cognitive constraint, i.e., our reliance on well-defined symbolic entities for understanding, has made deciphering the complexity of billions of neural activities an especially daunting task. It underscores a fundamental trade-off between the expressiveness of a model and our ability to understand it [86].

Focusing solely on individual neurons is also insufficient to capture the broader mechanisms underlying neural activity across a network. *Monosemantic* neurons, which respond to a single concept, make up only a small fraction of the overall neural population [72, 25, 85, 19, 84, 59]. Empirically, especially for transformer models [25], neurons are often observed to be *polysemantic*, i.e. associated

with multiple, unrelated concepts [63, 25, 67], which complicates the task of understanding how neural population activity evolves across layers [25, 39]. This highlights the need for more holistic approaches that account for the complex, distributed nature of neural representations.

# 3 Chunks from Neural Embeddings (CNE)

We effortlessly perceive high-dimensional perceptual signals by segmenting them into recurring, meaningful patterns, i.e., chunks. These patterns reside in a subset of our perceptual dimensions as cohesive wholes [58, 48, 37, 35]. We employ this strategy to simplify the complexity of naturalistic data. As naturalistic data possess a compositional structure and contain rich regularities, isolating recurring entities as concepts is an effective way for an agent to decompose their observations into entities and their relations [89, 87].

Similar to humans, AI systems aim to understand reality from naturalistic data. The regularities in naturalistic data may drive converging representations in diverse AI models. Recent findings suggests that neural networks of different architectures, scales, and sizes learn representations that are remarkably similar [5, 61], even when trained on different data sources [52]. This phenomenon is more pronounced in larger, more robust models [6, 21, 47, 74], and across multiple data modalities [52, 61, 47, 74]. Recently, it has been hypothesized that such convergences reflect an underlying statistical model of reality [41, 28].

**The Reflection Hypothesis**: We posit that the underlying cause of these observations is that, like human cognition, neural networks reflect the structures and regularities present in their training data through their internal computations. Specifically, we hypothesize that a well-trained neural network should exhibit trajectories of neural activity that mirror the structure of the data.

Formally, denote the sequence as $S = (s^1, s^2, \ldots, s^n)$, indexed by $I = \{1, 2, 3, \cdots, n\}$. A pattern in the input is defined as a recurring subsequence carrying the same concept $P = (s^i, s^{i+1}, \ldots, s^{i+k-1}) \subseteq S$ of fixed or variable length $k$. A neural network with a set of neurons $W$ ($|W| = d$) exerts a sequence of population activities $S_h = (\mathbf{h}^1, \mathbf{h}^2, \cdots, \mathbf{h}^n)$ in response to the input sequence. Each neural population vector has embedding dimension $d$: $\mathbf{h}^i \in \mathbb{R}^d$.

The Reflection Hypothesis posits that such recurring input patterns $P$ are mirrored within the internal neural dynamics of the model, such that their corresponding neural activations $(\mathbf{h}^i, \mathbf{h}^{i+1}, \ldots, \mathbf{h}^{i+k-1})$ lie within a population-level *chunk* defined as a ball $\overline{B}(\overline{\mathbf{h}}_C, \Delta) \subset \mathbb{R}^d$ centered at a prototypical activation vector $\overline{\mathbf{h}}_C$ with radius $\Delta$. Regularities in the input sequence are reflected as localized activity patterns in subdimensions of neural population dynamics.

If this hypothesis is even partially true, it suggests that neural network activity can be decomposed into recurring "chunks," analogous to how human perception segments high-dimensional visual input into meaningful units. Such chunks, elicited by activities of neural subpopulations, may represent distinct computations and serve as interpretable entities. Extracting these entities allows a reduction of the highly complex neural computation to the emergence of these chunks. We develop architecturally agnostic methods to extract chunks and evaluate their efficacy in concept encoding, interpretability, and their causal role in altering network behavior.

## 3.1 Extracting Entities

Depending on the dimensionality $d$ and the nature of the problem, we develop three chunk extraction methods: **discrete sequence chunking** for small $d$, **neural population averaging** for large $d$ and when there is an identifiable pattern in $S$, and **unsupervised chunk discovery** when the supervising concept is unavailable.

**Discrete Sequence Chunking (DSC)**   When $d$ is relatively small, we can use a cognitive-inspired method to extract recurring patterns [87], which is also used for text compression [31, 92, 2] or tokenization. The idea is to cluster the neuron activations individually, use the cluster indices to represent the network activity as strings, and then further group frequently occurring string sequences into chunks. For this purpose, we convert each hidden neural activity vector $\mathbf{h} \in \mathbb{R}^d$ into a string of discrete integers, and transform the sequence of vectors $S_h$ into a one-dimensional sequence of strings $S^s = (l^1, l^2, ..., l^n)$. Each $l^j$ has length $d$ and contains the nearest cluster index for each of the

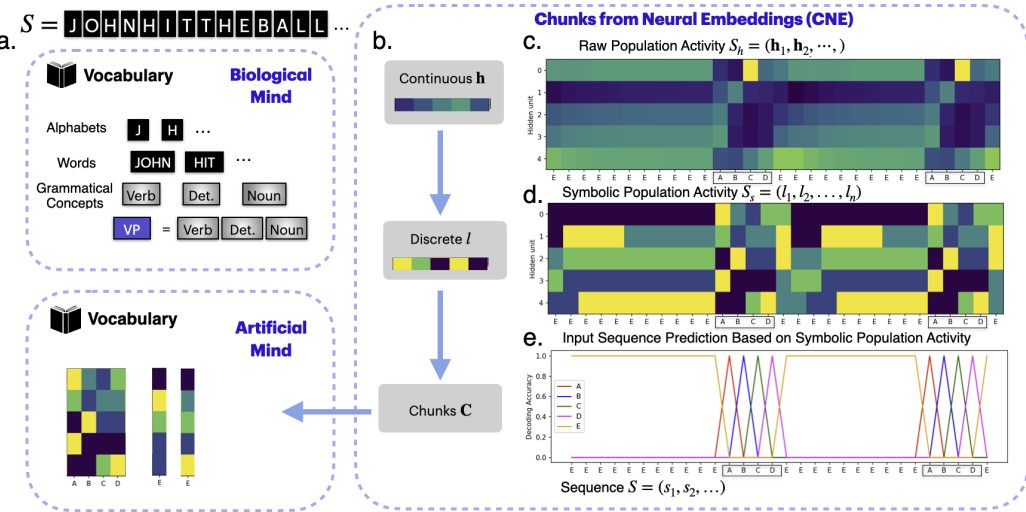

Figure 1: a. Naturalistic data are highly redundant and compositional, e.g. in language sequences, cognitive systems learn a dictionary of recurring concrete and abstract entities. b - e. In simple networks that contain a small number of neurons, chunking methods can be used to learn a dictionary of frequently recurring population trajectories. This discrete representation can reliably predict the input in the sequence and the network's predictions on the next character.

$d$ neurons, i.e., $l^j = \text{concat}(\mathcal{M}(h_1), \mathcal{M}(h_2), \dots, \mathcal{M}(h_d))$. $\mathcal{M}$ is a clustering function that assigns the closest cluster index to the activity of the neuron.

We then apply a chunking method on $S^s$ to extract a vocabulary of patterns of string combinations, which represent patterns in neural state trajectories. The vocabulary of chunks $\mathbf{D}$ is initialized with the set of unique strings in $S^s$. This vocabulary grows iteratively by merging the top $N$ frequent adjacent chunk pairs above a threshold. Within each new iteration, the updated vocabulary is used to reparse the sequence, repeating until convergence (see SI 10). This identifies recurring neural population trajectories that segment the activity sequence. The resulting dictionary $\mathbf{D}$ can then parse $S^s$ in terms of extracted chunks.

**Neural Population Averaging (PA)**   In larger-scale networks like transformers, $d$ can be very big. In this case, identifying recurring patterns in the sequence allows us to uncover the internal structures of neural population chunks by averaging the relevant population activities, akin to extracting task-induced neural response functions [17, 27, 16].

We assume that there are recurring activities of neural subpopulations, which account for the network's computation as elicited by a recurring concept $s$ in the sequence (such as a particular word 'cheese'). Denote the indices of the concept encoding chunks as $C(s) \subseteq W$, i.e., a subset of the neurons in $\mathbf{h}$ among the whole neuron population $W$ ($|W| = d$). Denote the set of indices where the pattern appears in the input sequence as $V(s) \subset I$. From the Strong Law of Large Numbers (SLLN), the population mean of concept-encoding neural subpopulation activities $\mathbf{h}_{C(s)}$ converges to the true mean as $|V(s)|$ approaches infinity:

$$\overline{\mathbf{h}_{C(s)}} = \frac{\sum_{j \in V(s)} \mathbf{h}_{C(s),j}}{|V(s)|} \text{ and } \lim_{|V(s)| \to \infty} \overline{\mathbf{h}_{C(s)}} = \boldsymbol{\mu} \quad \text{(a.s.)} \tag{1}$$

where $\boldsymbol{\mu} = \mathbb{E}[\mathbf{h}_{C(s)}]$ is the true mean of the subpopulation neural activity.

Given training data containing the sequence $S$, the corresponding neural population activity $S_h$, and a recurring pattern $s$ in $S$, we aim to estimate the signal-relevant chunk. This includes the mean neural subpopulation activity $\overline{\mathbf{h}_{C(s)}}$ within the subpopulation neurons $C(s) \subset W$, as well as the range of deviation $\Delta$ within which the neural activities $\mathbf{h}_{C(s)}$ fluctuate $\overline{\mathbf{h}_{C(s)}}$.

To do this, we first compute the mean population response of signal $s$ by averaging over the token-specific hidden state representations $\overline{\mathbf{h}} = \frac{\sum_{j \in V(s)} \mathbf{h}_{(j)}}{|V(s)|}$. A neuron $i$ is hypothesized to be inside $C(s)$ if its activity at the time of the concept-specific index fluctuates within a pre-set tolerance level around the mean signal-relevant activity $C(s) = \{i \in W : |h_{i,j} - \overline{h_i}| \le \text{tol} \quad \forall j \in V(s)\}$

After estimating the neural subpopulation $C(s)$ and $\overline{\mathbf{h}_{C(s)}}$, we then calculate a maximal deviation acceptable in the training data:

$$\Delta = \max_{j \in V(s)} \frac{||\mathbf{h}_{C(s),j} - \overline{\mathbf{h}_{C(s)}}||_2^2}{d} \tag{2}$$

We define the deviation $\Delta$ as a threshold for acceptable fluctuation around the mean neural subpopulation activity $\overline{\mathbf{h}_{C(s)}}$. An unknown neural activity vector $\mathbf{h}$ is classified as belonging to the chunk $\overline{\mathbf{h}_{C(s)}}$ if it lies within the closed ball $\overline{B}$ centered at $\overline{\mathbf{h}_{C(s)}}$ with radius $\Delta$:

$$\mathbf{h} \in \overline{B}(\overline{\mathbf{h}_{C(s)}}, \Delta) \quad \text{iff} \quad \frac{||\mathbf{h}_{C(s)} - \overline{\mathbf{h}_{C(s)}}||_2^2}{d} \le \Delta$$

Using this criterion, we identify instances of an unknown neural subpopulation vector $\mathbf{h}_{C(s)}$ in the test data using the chunk identification function

$$f_{\text{chunk}}(\mathbf{h}_{C(s)}, \overline{\mathbf{h}_{C(s)}}, \Delta(s)) = \begin{cases} 1, & \text{if } \frac{||\mathbf{h}_{C(s)} - \overline{\mathbf{h}_{C(s)}}||_2^2}{d} \le \Delta(s), \\ 0, & \text{otherwise.} \end{cases} \tag{3}$$

and evaluate its quality as a neural population activity classifier for the occurrence of the signal $s$ using true positive rate (TPR) and false positive rate (FPR).

Since the neural subpopulation $C(s)$, the chunk $\overline{\mathbf{h}_{C(s)}}$, and the deviation threshold $\Delta$ depend on a tolerance parameter, we generate a series of increasingly stringent tolerance thresholds: $\text{tol}_i = 2 \times 0.8^i$, $i = 0, 1, \ldots, 39$. As $\Delta$ is tied to the tolerance threshold, we find the optimal tolerance parameter and the associated $\Delta$ that maximizes TPR while minimizing FPR on the training data.

**Unsupervised Chunk Discovery (UCD)** While discrete sequence chunking applies to small $d$ and population averaging hinges on knowing the location of the recurring concepts, we also develop an unsupervised method to discover recurring patterns in the embedding space.

To this purpose, we formulate the chunk finding question as learning a dictionary matrix $\mathbf{D} \in \mathbb{R}^{K \times d}$ that includes the recurring chunks embedded in the latent space of dimension $d$.

Given the embedding data $\mathbf{X} \in \mathbb{R}^{M \times d}$ from one single layer of an LLM - processing some tokenized input with batch size $M$ and embedding dimension $d$, we optimize a loss function $\mathcal{L}(\mathbf{X}, \mathbf{D})$ that encourages each embedding to match its most similar dictionary entry:

$$\mathcal{L} = -\frac{1}{M} \sum_{m \in \{1, \cdots M\}} \max_{k \in \{1, \ldots, K\}} \mathbf{SIM}(\mathbf{D}_k, \mathbf{X}_m) \tag{4}$$

Here, $\mathbf{SIM}(\mathbf{d}, \mathbf{x})$ denotes the normalized cosine similarity between $\mathbf{d}$ and $\mathbf{x}$: $\mathbf{SIM}(\mathbf{d}, \mathbf{x}) = \frac{\mathbf{d}^\top \mathbf{x}}{||\mathbf{d}||_2 ||\mathbf{x}||_2}$. The loss function encourages each embedding $\mathbf{X}_m$ to align closely with one of the chunks $\mathbf{D}_k$.

## 4 Results

### 4.1 Evaluating Reflection Hypothesis on Simple RNNs

First, we assessed the reflection hypothesis using a simple recurrent neural network (Figure 2a and appendix D.1) trained on synthetic sequences containing a known pattern. We begin with sequences in which ABCD appear periodically (Figure 2b), then gradually increase complexity: first by embedding ABCD sparsely within a background of repeated Es (Figure 2c), and then by placing ABCD amid randomly occurring background symbols (E, F, G) to simulate noise (Figure 2d). In each case, the RNN is trained to predict the next character. Across all conditions, the network's hidden states consistently exhibit activity patterns that reflect the pattern occurrences in the sequence.

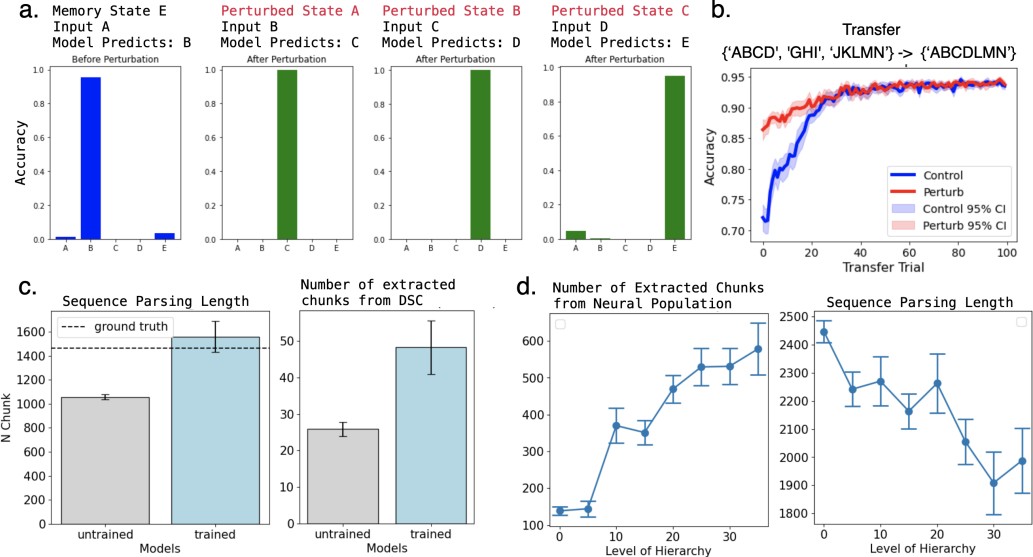

Figure 2: a. Hidden states can be grafted to causally change network memory and prediction. b. Embedding grafting enables faster transfer learning of a compositional vocabulary. c. When RNNs learn from sequences that contain context-dependent predictions, training creates extra chunks inside the embedding space. d. Left: The neural population trajectory can be parsed by bigger chunks, which translates to smaller sequence parsing length with an increasing level of hierarchy in the training sequence. The number of extracted chunks increases with the level of hierarchy inside sequences.

**Understanding the neural state and its mapping to the input**  We apply *Discrete Sequence Chunking* to transform continuous hidden unit activities to a symbolic and discretized description. We denote population activities by the alternation of indices that mark the belonging cluster centroid of individual neurons and visualize the population activity by the indices of the assigned nearest cluster. This symbolic description of the neural activities allows decoding the trajectory of the hidden state and its corresponding input via a look-up table (appendix D.2), and reaches a perfect decoding accuracy on the test set. From the symbolic description of neural trajectories, we can then apply chunk discovery methods to learn a dictionary of symbolized chunks. This dictionary contains the maximally recurring patterns inside the network, which consistently reflect the patterns in the input.

**Grafting PA chunks causally alter the network's behavior**  We then test whether the extracted chunks causally alter the network's output in a predictable way. Starting from a state predicting B from input A (with prior input E), we replace the hidden state with those corresponding to A, B, or C, while pairing with inputs B, C, or D, reliably shifts the output predictably to C, D, or E (Figure 2 a).

**Grafting neural population activity in chunks facilitates compositional learning**  Humans learn complex sequences by reusing simpler parts [51, 50]. RNNs, by contrast, struggle with reuse and composing the new from the known [49]. We test whether the population grafting may artificially induce the network to compose and reuse a previously learned vocabulary to compose a new vocabulary. We trained two identical RNNs on synthetic training sequences with randomly sampled words from a dictionary {ABCD, GHI, JKLMN} occurring within null E characters.

We then further train the two RNNs in a transfer sequence with the word ABCDLMN - a composite of two-word parts in the training vocabulary. Without perturbation, the control RNN suffers from a performance degradation consistent with the observations in the literature. For the other RNN, we learn a lookup table that maps its discrete population state to the concurrent input. As this RNN learns on the transfer sequence, whenever the input is D (the network state should be C), we graft the neural population state to the cluster centroid of the hidden state induced by the previous input being J paired with the input character being K, thereby forcing the network to generate the next prediction as L instead of E. Shown in Figure 2 b are the learning curves between the two networks during the transfer sequence. The perturbed RNN predicts the transfer sequence with higher accuracy than

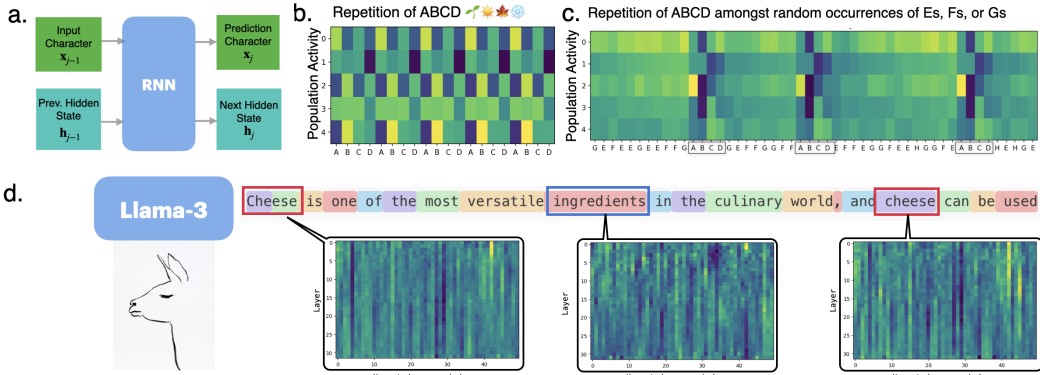

Figure 3: *The reflection hypothesis*: ANNs' neural activities can be parsed into entities that reflect the structured regularities in training sequences. a-c. Population activity of the initial 5 neurons (unsorted) from an RNN trained to predict the next sequence character, from a sequence that contains repeating ABCD patterns and another that contains ABCD embedding in a default background noise (random E, F, and G); d. Raw neural activity of the first 50 neurons of LLaMA-3 (unsorted) across all layers (32) processing the prompt up to the last token of each highlighted word.

control, artificially grafting hidden states forcefully induces the RNN to reuse existing representations to predict a sequence composition.

**Training creates additional population states that distinguishes contexts**   We further hypothesized that learning may create additional patterns of neural population activities inside the network to distinguish different contexts. We create sequences with a vocabulary that contains context-dependent subsequence parts: a vocabulary {CDAB, AB, ABCD} among a default sequence of Es. In this sequence, the CD inside ABCD conveys a different meaning than that inside CDAB. While the former predicts E shall follow, the latter predicts that AB shall follow. We used DSC to learn a dictionary of chunks from an untrained and a trained network separately, responding to the input sequence of a fixed length. We then use the learned chunk to parse the neural trajectory and measure the length of the parsing. Shown in Figure 2 c, a trained RNN contains a neural trajectory that can be parsed by a number of chunks similar to the number of words in the sequence, and is higher than the untrained network. The trained network also processes neural trajectories that contain a more diverse set of chunks, suggesting that training creates additional neural trajectory patterns to distinguish context-specific meanings.

**Hierarchically structured input sequence corresponds to a more diverse number of chunks in neural activities.**   Another implication of the reflection hypothesis is that when the input sequence contains an inherent hierarchical structure, i.e., alphabets can become word parts, which can become words, etc., then a neural network that learns to predict the sequence should also exhibit a more diverse set of population state to distinguish the ever-more complex context dependency given rise by the hierarchical structure. To test this, we generated synthetic sequences with hierarchical vocabulary following [87]. Starting with an alphabet {A, B, C, D} and default null symbol E, we iteratively built new words by concatenating pairs from the existing vocabulary over $c$ steps. Word probabilities were sampled from a uniform Dirichlet distribution, scaled by 0.2, with the remaining 0.8 assigned to E. We sampled from this vocabulary to construct the training sequences. We then independently trained RNNs on sequences with an increasing number of hierarchical depth $c$ and extracted neural population chunks using DSC. As shown in Figure 2d, the number of extracted neural chunks grew with hierarchy level/vocabulary size. Additionally, as the structure of the input becomes more nested, population trajectories could be parsed into increasingly larger chunks of neural trajectories, indicating that the network internalized the nested structure.

### 4.2   Reflection Hypothesis Evaluated on Large Language Models

We next test the reflection hypothesis on larger models, and then use the population averaging and unsupervised chunk learning method to extract recurring entities that may reflect the network's

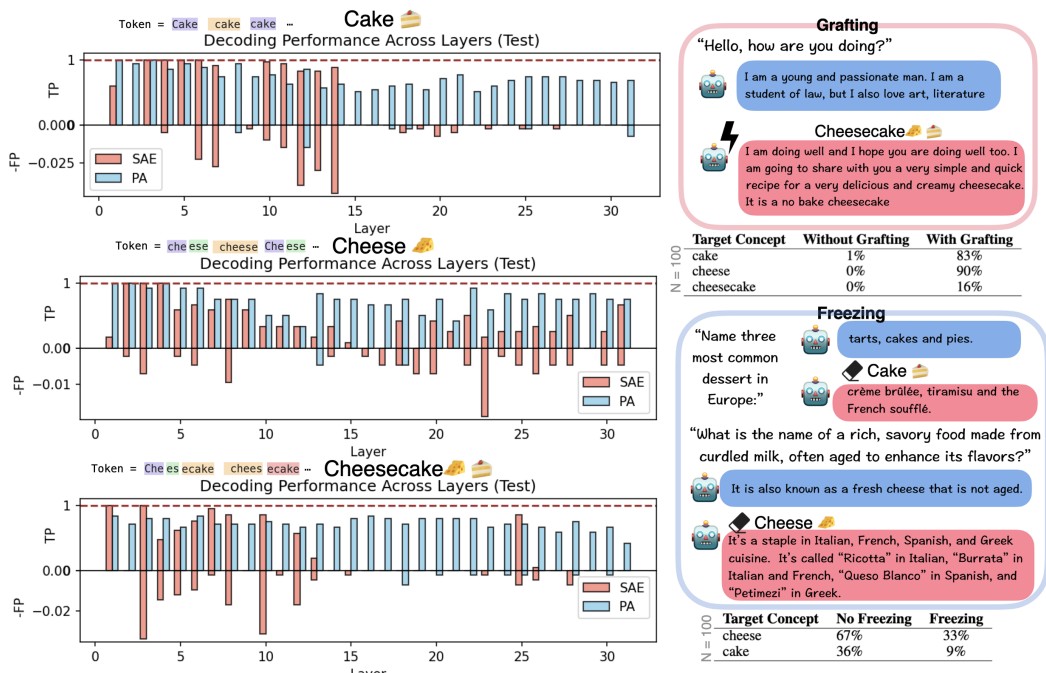

Figure 4: The identifiability of the presence of extracted chunks evaluated by signal detection measures, and a comparison between SAE. Grafting and freezing word-related population chunks alter the network's sequence generation.

representation of concepts. We started with LLaMA3-8B [22] due to its open-source access and architectural complexity. LLaMA3 predicts the next token based on the ever-expanding preceding context; hence, it analyzes LLaMA3's hidden states as it incrementally processes tokenized sequences, from the beginning of each sentence up to the latest $n$th token $S = (s^1, s^2, ..., s^n)$. We recorded the hidden state sequence across all layers $S_h = (\mathbf{h}^1, \mathbf{h}^2, ..., \mathbf{h}^n)$ to track the evolution of the model's internal activity as new tokens are included. We then examine the existence of neural chunks applying the *Neural Population Averaging* and *Unsupervised Chunk Discovery*.

**Recurring concepts in a sentence** Figure 3d shows normalized activations of the first 50 neurons (unsorted) across all LLaMA3's layers for the word "cheese". Consistent with the reflection hypothesis, the two differently tokenized occurrences of "cheese" evoke similar neural activity patterns, more so than a different word like "ingredient," suggesting the presence of common neural subpopulation activities encoding recurring concepts. Using recurring words in language as the pattern of interest $s$, we applied PA to identify the neural subpopulation $C(s)$, the mean chunk activity $\overline{\mathbf{h}}_{C(s)}$, and the fluctuation radius $\Delta(s)$ at the last token of each word bearing a variety of tokenization forms.

We assessed the extracted chunks and thresholds by measuring how well chunk detection predicted the occurrence of the word concept $s$. Figure 4 shows that the population averaging method yielded neural

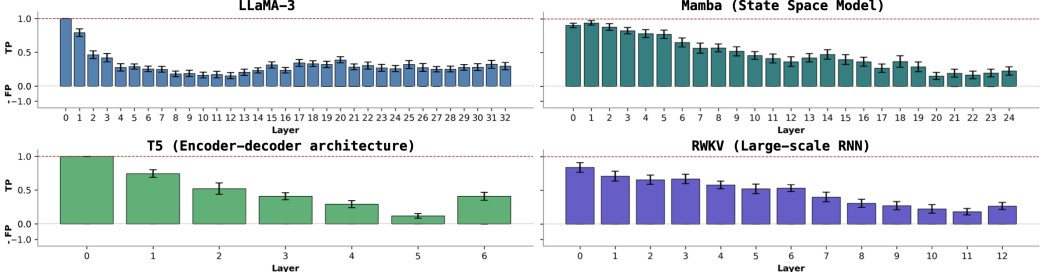

Figure 5: Average decoding performance of the top 100 words in English, applying population averaging on a variety of models. Error bar denotes the standard error of the mean.

subpopulation chunks that are predictive of the example words "cheese", "cake", and "cheesecake" in each layer and distinct from one another, with low false positives and high true positives across all layers throughout. Additionally, we compared PA with a pretrained sparse autoencoder (SAE) on LLaMA-3 [24], using the most active concept-related neurons as decoders. Similar to PA, we selected the activation thresholds of concept encoding neurons from training data and applied them to test data. Overall, PA-extracted population activity predicts the presence of the selected concept better than the individual concept neurons identified by SAE.

We generalized our method for identifying concept-encoding neural population chunks to other large-scale models with distinct architectures and sequence-processing mechanisms, including the encoder-decoder model T5 (t5-small [66]), the RNN-based RWKV (rwkv-4-169m-pile) [70], and the state-space model Mamba (mamba-130m-hf [38]; see Figures 24, 25, 26 in the SI for decoding the example concepts). We evaluated chunk quality by testing whether concepts could be predicted from chunk detections on held-out neural recordings corresponding to test prompts. Beyond the illustrated concepts, we extended the concepts to the top 100 most frequent English words. Overall, we found that PA consistently extracts chunks whose activations correspond to meaningful concepts across diverse architectures. The presence of a chunk reliably predicts the presence of its associated concept in the text with minimal false positives.

**Chunks Encoding Abstract Schema and Narrative Structures**    We evaluated PA's ability to find chunks encoding more abstract narrative structure by training on 20 stories following a schema (e.g., visit food location → buy item → eat → react) and 13 control stories with narrative structure inconsistent with the schema (examples shown in Figure 6). We extracted shared subpopulation chunks at the end-of-sentence token using PA, and tested on 18 new schema-consistent and 15 inconsistent stories. Schema-related chunks are activated at sentence boundaries in structured narratives but not in controls, suggesting that PA captures abstract narrative schemas beyond surface-level concepts or syntax.

**Grafting and Freezing Neural Populations**    We tested the causal roles of these extracted signal-relevant neural subpopulation activities $\overline{\mathbf{h}_C(s)}$ by grafting neurons to discovered chunks and generating the subsequent text with LLaMA-3. To do this, we fed in a prompt and graft the neural subpopulation $C(s)$ to $\overline{\mathbf{h}_C(s)}$ at a specific token position of the prompt. Shown in Figure 4, perturbing the hidden units to embeddings corresponding to words such as "cheese", "cake", or "cheesecake" biases the network towards generating sequences related to the grafted topic. Conversely, we also experimented with freezing the words by setting the corresponding neural subpopulation of chunk support set to zero $\mathbf{h}_{C(s)} = \mathbf{0}$, and used prompts that lure the network to generate the frozen word concept. As shown in Figure 4, freezing the chunk related to a topic can cause the model to avoid using the concept-specific word.

We then evaluate the general effectiveness of grafting general concepts on the ROCStories benchmark [62], sampling 2,000 sentences to extract neural population chunks corresponding to the top 20 frequent concept words (filtered to avoid short or overlapping terms) in addition to the existing concepts collected. To assess context-dependent effects of prompt used when grafting the concept-encoding neural chunks, we used prompts from the TREC Question Classification dataset [53], which includes six coarse-grained context categories. For each category, we sampled 50 prompts and evaluated grafting success by measuring the difference in concept occurrence probability between grafted and ungrafted (control) LLaMA-3 generations. To reduce computational cost, we categorized grafting into early (layers 1–9), middle (10–19), and late (20–29) interventions. Results are shown in Table 1. We found that grafting consistently increased concept occurrence across contexts, with early-layer grafting being the most effective.

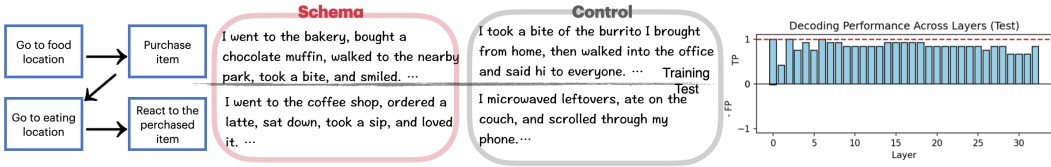

Figure 6: Example sentences consistent with one narrative structure, and the control sentences which are inconsistent in training and test sets. Decoding accuracy of narrative-consistent structures in the test set.

Table 1: Effect of grafting on TREC categories (percentages).

| TREC Category | No Graf. | Early Graf. | Middle Graf. | Late Graf. |
|---|---|---|---|---|
| ABBR (Abbreviations, acronyms) | 14.9% | 55.9% | 30.8% | 18.0% |
| DESC (Descriptions, definitions) | 15.6% | 49.0% | 28.1% | 20.7% |
| ENTY (Entities) | 12.6% | 48.1% | 22.5% | 16.9% |
| HUM (Human-related) | 11.9% | 46.7% | 21.5% | 15.2% |
| LOC (Locations) | 10.7% | 47.5% | 20.5% | 14.4% |
| NUM (Numeric answers) | 11.5% | 45.3% | 21.8% | 16.0% |

**Unsupervised Chunk Discovery**   We investigate whether recurring chunks in the neural embedding space can be identified in an unsupervised manner, when labels in text are unavailable. To this end, we train a chunk dictionary $\mathbf{D}$ on LLaMA3's hidden activity while processing sentence-wise prompts derived from *Emma* by Jane Austen, sourced from the Project Gutenberg corpus [40], accessed via NLTK [11]. As this is a bigger embedding dataset, we then trained $\mathbf{D}$ ($K = 2000$, $d = 4096$) by minimizing the similarity loss function formulated in 3.1 individually for each layer of LLaMA-3.

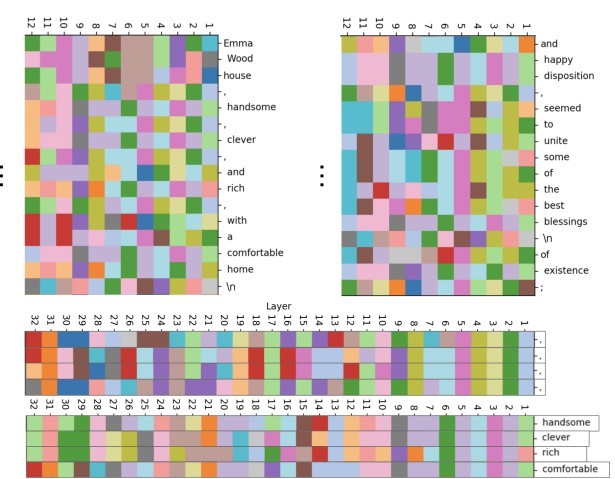

Figure 7: Upper: Describing the network's processing of tokens as the succession of unsupervised learned chunks. Lower: Layer-wise processing of similar token types.

This way, we can simplify high-dimensional embeddings by representing them as the appearance and disappearance of identifiable chunks. This enables visualization of the interactions between input tokens and the activations of these recurring entities. Figure 7a illustrates chunk interactions across LLaMA's layers during early text processing. Recurring chunk patterns emerge. For example, commas and adjectives show similar activations in early layers (Figure 7b), but their representations later elicit distinct chunks in deeper layers. The full plot can be found shown in SI F.5. We also interpret the unsupervised learned chunks by correlating them with part-of-speech (POS) tags, the result can be found in SI F.6.

## 5   Conclusion, Discussion

We propose the *Reflection Hypothesis*, which posits that neural population activity reflects the structure of the data. We provide evidence in support of this hypothesis and, building on this insight, leverage the cognitive principle of chunking to isolate perceptual entities as units of interpretation. Using the methods we introduce, we identify chunks and test their feasibility across models ranging from small-scale RNNs to a number of large-scale LLMs. We find that these chunks correlate strongly with both concrete and abstract concepts—spanning recurring words, sentence structure, and POS tags. These results suggest that chunking provides a powerful lens for segmenting high-dimensional neural activity into structured trajectories. This work invites future research to formalize its theoretical foundations, investigate the learning dynamics that give rise to such entities, and apply this perspective to build a mechanistic understanding of the inner workings of complex learning systems.

## 6   Acknowledgment

This work was partially funded by the ERC (853489 - DEXIM) and the Alfried Krupp von Bohlen und Halbach Foundation, for which we thank them for their generous support. Shuchen Wu was in part supported by the Shanahan Family Foundation Fellowship at the Interface of Data and Neuroscience at the Allen Institute and the University of Washington. Stephan Alaniz is in part supported by Hi!

PARIS and ANR/France 2030 program (ANR-23-IACL-0005). Peter Dayan is funded by the Max Planck Society and the Humboldt Foundation. Shyamgopal Karthik thanks the International Max Planck Research School for Intelligent Systems (IMPRS-IS) for support.

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

## Supplementary Information (SI)

Implementation and code are publicly available at `https://github.com/swu32/Chunk-Interpretability`.

## A    Extended Discussion on Related Work

### A.1    Comparison with Other Interpretability Approaches

Our work is a different approach from a number of other interpretability methods. For example, Black et al. define polytopes as linear regions in input space and study their geometry [12]. Vielhaben et al. define concepts as linear subspaces in feature space, spanned by an unbounded number of basis vectors [83]. Our definition of a chunk is compact and bounded subspace, hence it does not tolerate addition or scalar multiplication of some basis vectors. In fact we would expect a certain level of scalar multiplication may run into other chunks. We suggest that chunks themselves are interpretable.

Other approaches, such as Tamkin et al., propose architectures that enforce sparsity and interpretability. In particular, they learn a codebook of features through codebook bottlenecks [78], which requires training new models with modified architectures. By contrast, our goal is to interpret the raw activity of large models directly, independent of their training procedures or architectures, rather than optimizing a new model.

### A.2    Comparison with SAEs and Representation Engineering

Our method for grafting and freezing corresponds to steering and unlearning in SAE work [29, 57]. However, the chunk grafting is simpler and more direct than that of SAEs. In SAEs, steering is typically achieved by clamping a feature to a specific value—either through a constant multiplier or a scaled value relative to its maximum activation [81]. This clamped feature then probabilistically influences the network's output during text generation.

However, SAE features often do not correspond clearly to human-interpretable concepts. The mapping between features and words is indirect, requiring additional steps such as VocabProj or MaxAct to identify associated tokens [29]. This makes steering both complex and fragile. Interpreting SAE features remains an open challenge, and developing effective steering methods is still an active area of research [77].

In contrast, PA offers more interpretable units by design. Chunks are derived from recurring patterns in the model's activations, grounded in actual input semantics. This enables more transparent and effective interventions, such as concept grafting or unlearning, without the need for specialized heuristics. We also evaluated our steering efficiency on a bigger corpus across multiple contexts, and verified that the extracted chunks can be an efficient way of steering network behavior.

The other main difference between this approach and SAE is compute efficiency. The population averaging (PA) method is a post hoc analysis that requires no optimization or gradient computation. Neural activations are precomputed, and the only cost lies in computing prototype vectors and evaluating distances across a small number of deviation thresholds—operations that scale linearly with the number of tokens and the embedding dimension.

In contrast, sparse autoencoders (SAEs) require multiple forward and backward passes through both encoder and decoder networks, typically over many training epochs. This results in orders-of-magnitude greater computational cost. PA, by contrast, offers a lightweight and accessible alternative for large-scale interpretability research. The unsupervised chunk discovery method also remains computationally feasible. It runs efficiently on a single GPU. Given precomputed neural activations, it can be applied broadly to various architectures and datasets.

This work also differs from representation engineering, which isolates neural activity associated with a predefined concept or function, such as bias or truthfulness [94]. Representation engineering typically fits a linear model to identify activity directions that predict or intervene on the target concept. However, this approach assumes the concept of interest is known a priori. In contrast, we assign a single chunk to summarize the neural activity of a concept, enabling discovery without prior labels. In contrast to grafting with chunks, which directly overwrites neural activity with a given chunk, representation engineering applies a latent direction whose effect depends on a chosen magnitude. Determining this strength is a nontrivial hyperparameter choice and not always straightforward.

### A.3 Comparison with Concepted-based Interpretability Literature on Vision Models

Our work is different from concept-based interpretability methods for image classifiers. In [34], [30] and [45] concepts correspond to visual features in the image dataset (e.g. pixels of a wheel), and they study what part of the image influences the network's classification decision. In contrast, our work primarily studies network interpretability processing sequences of tokens. Our work examines how the internal structure of the network exhibits recurring patterns of entities. We reframe interpretability by connecting neural population trajectories to human chunking.

## B  Definition of Grafting and Freezing

**Grafting** refers to the targeted insertion of previously learned chunks into a neural network's internal computation. We modify its internal activations $\mathbf{h}$ at a specified layer $l$ and neural subset $C$ by grafting the chunk $\mathbf{h}_{C,l} \leftarrow \overline{\mathbf{h}_{C,l}}$ to replace the original neural subpopulation activities.
**Freezing** refers to the operation of zeroing out the neural activations that support the chunk. This is implemented by replacing the network activation with stored chunk representations at the corresponding neurons. At a given layer $l$, we enforce: $\mathbf{h}_{C,l} = \mathbf{0}$.

## C  Pseudocode of Learning Chunks Using Discrete Sequence Chunking

---

**Algorithm 1:** LearnChunks

    **Input**   : $K$, $freq\_threshold$, $symbolized\_neural\_states$, $n\_iter$
    **Output**: $state\_parse$, $vocab$

1   $vocab \leftarrow$ unique($symbolized\_neural\_states$);
2   $null\_state \leftarrow$ GetMostFreq($symbolized\_neural\_states$);
3   $state\_parse \leftarrow$ Parse state trajectory by individual units;
4   **for** $n\_iter$ **do**
5      $ChunkCandidates \leftarrow$
        MostCommon($K$,Counter(zip($state\_parse[:-1]$, $state\_parse[1:]$)));
6      $merged\_dict \leftarrow$ Merge($c_L, c_R$) $\in ChunkCandidates$ if (count($c_L, c_R$) $\geq$
        $freq\_threshold$ and $c_L \neq null\_state$ $c_R \neq null\_state$);
7      $vocab$.update($merged\_dict$);
8      MergeOverlappingChunks($vocab$);
9      $state\_parse$, $vocab \leftarrow$ ParseStateSeq($symbolized\_neural\_states$,$vocab$);
10 **return** $state\_parse$, $vocab$;

---

---
**Algorithm 2:** Neural Population State Chunking
---
**Input** : $training sequence$
**Output** : $state\_parse$, $vocab$

**1** Initialize RNN;
**2** $hidden\_states$, $sequence \leftarrow$ TrainRNN($sequence$)
**3** $symbolic\_hidden\_states \leftarrow$ ClusterAndAssignSymbolToEachNeuron($hidden\_states$);
**4** $state\_parse$, $vocab \leftarrow$ **LearnChunks**($symbolic\_hidden\_activity$, $sequence$);
---

## D   RNN

### D.1   Recurrent Neural Network (RNN) Architecture

We provide more details on the structure of RNN. The RNN used by this work can be described by the following equation:

$$\mathbf{h}_n = \mathbf{W}_{ch} \begin{bmatrix} \mathbf{x}_n \\ \mathbf{h}_{n-1} \end{bmatrix} + \mathbf{b}_h \tag{5}$$

$$\mathbf{o}_n = \mathbf{W}_{co} \begin{bmatrix} \mathbf{x}_n \\ \mathbf{h}_n \end{bmatrix} + \mathbf{b}_o \tag{6}$$

$$y_t = \log\left(\text{softmax}(\mathbf{o}_t)\right) \tag{7}$$

The hidden state is initialized as all zeros $\mathbf{h}_0 = \mathbf{0}$. $\mathbf{x}_n, \mathbf{b}_h \in \mathbb{R}^{|\Omega|}$, $\mathbf{h}_n, \mathbf{b}_o \in \mathbb{R}^d$, $\mathbf{W}_{ch} \in \mathbb{R}^{d \times (d+|\Omega|)}$, $\mathbf{W}_{co} \in \mathbb{R}^{|\Omega| \times (d+|\Omega|)}$.

We implemented a simple RNN with 12 hidden units, consisting of two linear layers: one for updating the hidden state and another for generating output predictions. At each time step, the input vector is concatenated with the previous hidden state and passed through these layers. The output is normalized using log softmax to produce a probability distribution over the output classes. The RNN is optimized with cross-entropy loss using Adam (learning rate $= 0.005$). Training is conducted on random subsequences of length 200 per batch, with the hidden state initialized to zero at the start of training. We track hidden states for analysis.

For the experiment in Figure 1 of the main paper, the training sequence alternates between pattern ("ABCD") and noise ("E") repeated $k \sim \mathcal{U}(1, 20)$ times. The RNN processes one-hot inputs from a vocabulary, with inputs concatenated to hidden states and passed through two linear layers to produce log-probabilities. The model was trained for 160 iterations using Adam (lr = 0.005), with each iteration sampling a 200-character sequence for next-character prediction via cross-entropy loss.

To evaluate cross-model consistency, we applied the unsupervised chunk discovery method to RNNs with different architectures. All models exhibited consistent structural regularities (Figure 10), with similar chunk counts and alignment to input patterns (Figure 11) despite having different architectures.

### D.2   Example of a Lookup Table that maps from symbolic clustered state to the input

We apply *Discrete Sequence Chunking* to the hidden states of a simple RNN which was trained on a sequence of ABCD within default E characters. In table 2, we show the resulting lookup table for the discrete strings corresponding to input signals. Many distinct population states correspond to Es, and for A, B, C, and D, separately, there is a distinct population state for each. Presumably, the network creates additional states also to distinguish Es in different contexts.

### D.3   Validation of PA on RNNs

We also validated that PA works for simpler cases on RNNs demonstrated earlier in the paper by applying PA and UCD to RNNs trained on sequences with repeated "ABCD" patterns. PA revealed recurring chunk states whose activation frequency perfectly matched pattern occurrences in the sequence 8.

| Neural Population State | Input |
|:---:|:---:|
| 021340200433 | E |
| 004042212403 | E |
| 032340212204 | E |
| 032410212204 | E |
| 032010212204 | E |
| 221103343111 | A |
| 213211131132 | B |
| 144304324321 | C |
| 040322404444 | D |
| 300000000020 | E |
| 340432012022 | E |
| 440332412200 | E |
| 400040312200 | E |
| 402040312202 | E |
| 002040312202 | E |
| 032010212202 | E |

Table 2: Sample Look-up Table. The left column corresponds to the string that represents the neural population state. The right column are the corresponding concurrent input.

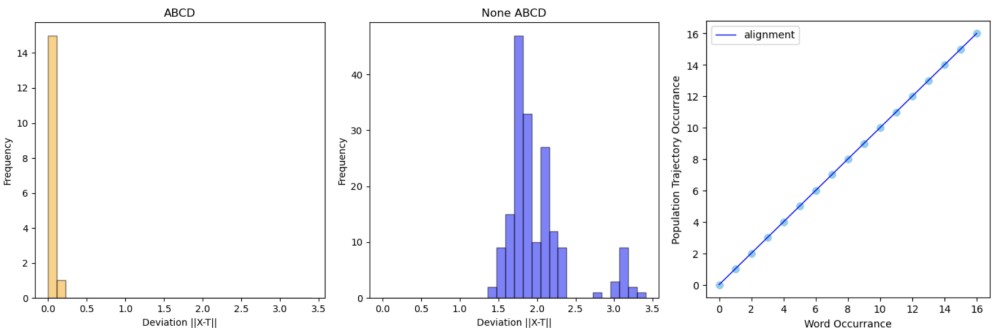

Figure 8: Applying population averaging on RNNs predicting simple sequences. Left: The chunk extracted from population averaging aligns closely with the signal-triggered neural embeddings. Middle: When ABCD does not occur in the sequence, embedding deviates strongly from the extracted chunks. The count of chunk occurrence in the neural population (x axis) correlates with the number of ABCD occurrences in the sequence. Blue line: perfect alignment.

We look at the neural population activity when the network learns from a sequence with repeating ABCD as a cohesive chunk amid background noise composed of random occurrences of E, F, and G. We get the population template by averaging the population activity vector $\mathbf{H} \in \mathbb{R}^{d \times 4}$ over the occurrences of the chunk ABCD:

$$\mathbf{T} = \frac{\sum_{i=1}^{L-4} \mathbb{1}\left((s_i, s_{i+1}, s_{i+2}, s_{i+3}) = \text{ABCD}\right) \mathbf{H}_i}{\sum_{i=1}^{L-4} \mathbb{1}\left((s_i, s_{i+1}, s_{i+2}, s_{i+3}) = \text{ABCD}\right)} \quad (8)$$

$$\text{dev} = \frac{\|\mathbf{H}[:, i:i+4] - \mathbf{T}\|_2^2}{\text{size}(\mathbf{T})} \quad (9)$$

Figure 8 suggests that neural activities, once adapted to the training data, deviate very little from the neural population template. Setting a threshold between the population activity and template deviation suffices to distinguish the occurrences of the input pattern based on the neural trajectory.

### D.4 Validate UCD on RNNs

To evaluate cross-model consistency, we applied the unsupervised chunk discovery method to the simple example in RNN. UCD learns similar chunks which aligns with the DSC extracted chunks (Figure 9).

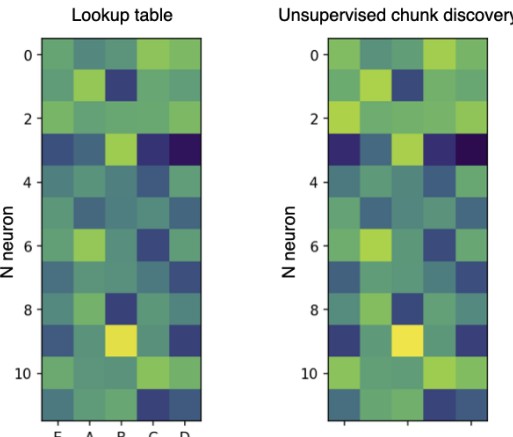

Figure 9: Applying the unsupervised chunking method to a simple sequence containing repeated instances of ABCD. Left: recurring population activity extracted from the discrete chunk learning. Right: recurring population activity extracted from the unsupervised chunk discovery method.

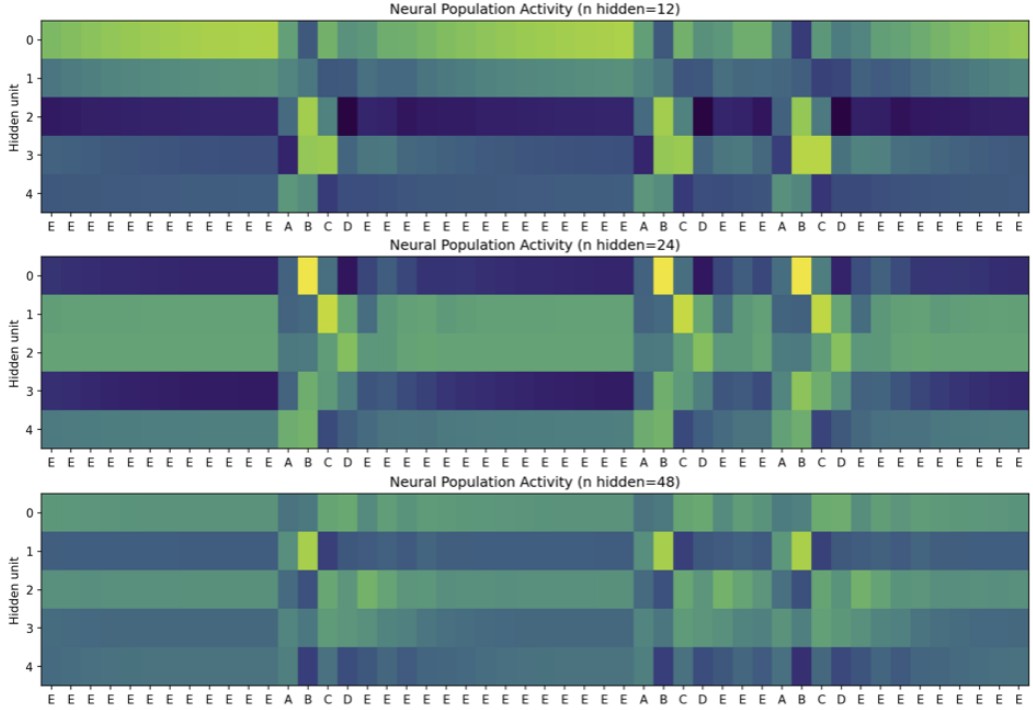

Figure 10: Neural population activity of the first 5 neurons (unsorted) across three distinct RNNs. Population activities exhibit a regularity reflecting patterns of the input sequence despite different network architectures.

### D.5 The Reflection Hypothesis Across RNN Architectures

We show that the reflection hypothesis stays intact, independent of the RNN's size. Figure 10 plots RNN's neural activity with an increasing number of hidden neurons (12, 24, and 48, separately). Across all RNNs, the neural activities reflect regularities in the training sequence.

Moreover, applying DSC on the RNNs suggests that the number of chunks extracted stays consistent across RNNs with different numbers of hidden units (Figure 11).

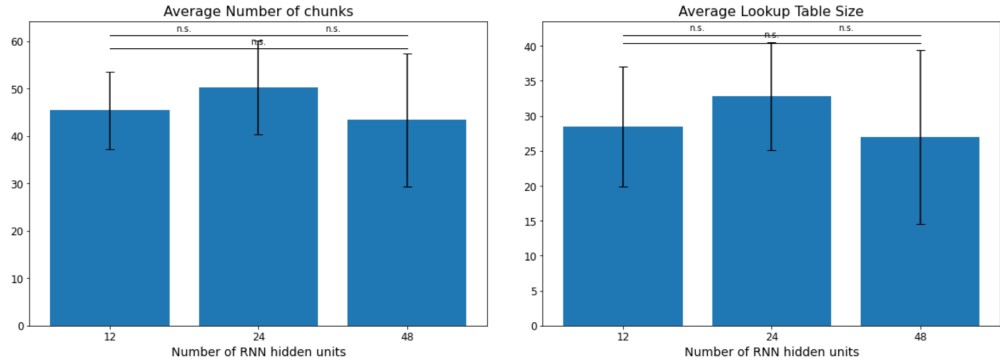

Figure 11: The number of chunks stays consistent across RNN with different number of hidden units.

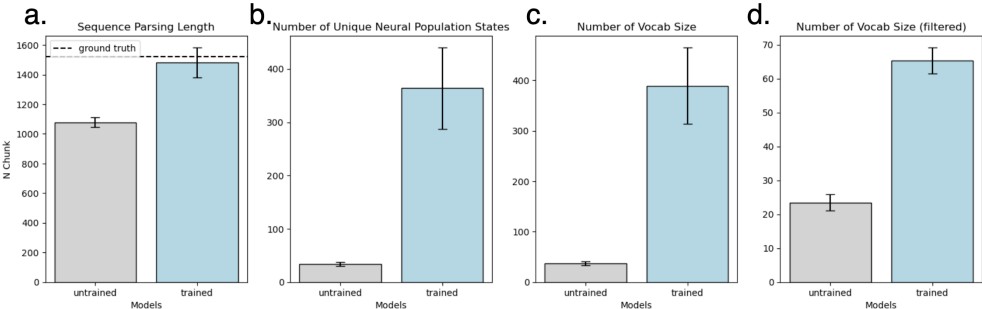

Figure 12: Comparison between trained and untrained RNNs on sequences with overlapping words across 10 independent runs and randomly initialized RNNs. a. Sequence parse length. Dashed line is the ground truth sequence parsing length. b. Number of unique neural population states. c. Chunk dictionary size. d. Chunk dictionary size (filtered desolate chunks (threshold=5)) Error bars represent the standard error of the mean.

## D.6 Contrasting Trained versus Untrained RNNs

We provide a comprehensive comparison of trained and untrained RNNs, focusing on the characteristics of their neural trajectory patterns as revealed by the chunking method.

Figure 12 a takes the neural population trajectory as a sequence, uses the learned neural chunks to parse the symbolic neural trajectory, and measures the length of the population trajectory sequence broken into chunks. A comparison of the sequence parsing length suggests that the trained network has a similar amount of neural trajectory chunks to the number of chunks in the ground truth sequence. The untrained network contains more regularities in its neural trajectory. This is an indicator training that encourages the network to create new distinct neural trajectory states that help the network distinguish different contexts. This property is especially important in this task as overlapping subsequences with different prefixes shall lead to distinct predictions.

Figure 12 b suggests that the trained RNN contains more unique neural population states than an untrained RNN. This observation supports the hypothesis that during training, RNN creates more population states that are useful for distinct contextual predictions. Figure 12 c, d, shows the vocabulary size as acquired by the symbolic chunking method. d is a filtered version of c, where obsoletely occurring chunks are excluded. The trained RNN's neural trajectory contains more recurring chunks than the untrained RNN. Also note that the size of the vocabulary containing recurring neural trajectories is bigger than the ground truth vocabulary in the sequence, pointing to a possibility that neural networks acquire multiple neural population chunks that can represent the same word in the sequence. However, the larger vocabulary size can also be an artifact of the clustering algorithm, which has more clusters than proper.

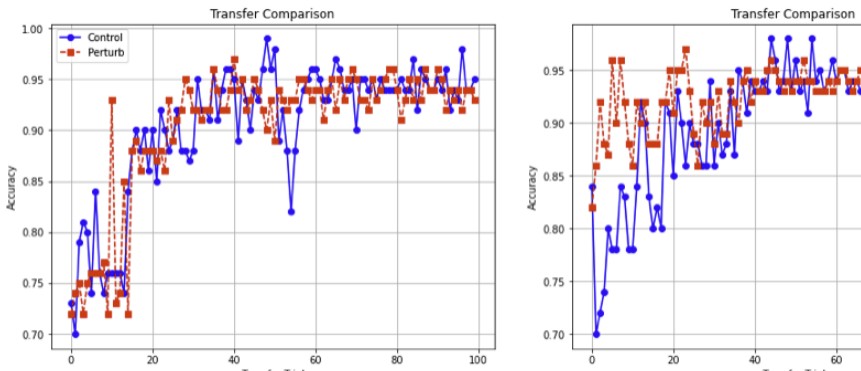

Figure 13: a. Simply grafting inputs is ineffective and do not encourage reuse. b. Grafting only the neural chunks encourages compositional reuse and transfer.

### D.7 Grafting Neural Chunks versus Grafting Input

We examine the effects of grafting at two levels—neural population chunks and input—on the behavior of RNNs when transferring to sequences with compositional components. Chunk-level grafting leads to reliable reuse (Fig. D.7); input grafting proves ineffective. This suggests that merely changing the input is insufficient to alter network behavior; rather, behavior is shaped by the coordinated activity of a larger set of computing units beyond those directly encoding the input.

## E Population Averaging

### E.1 Example Layer-wise Statistics PA-extracted Chunks

Beyond using PA to identify concept-encoding chunks for concepts occurring at the end of a prompt sequence, the same approach can also be applied to other positions in the sequence. Specifically, PA can isolate chunks that encode concepts appearing at earlier points in the sequence (prior to the end token), as well as chunks that encode the future occurrence of a target concept. The latter correspond to invariant neural subpopulations whose activity reliably precedes the later appearance of a given concept, which corresponds to predictive structure in the network's dynamics.

Figures 14, 15, 16, and 17 present layer-wise statistics of the learned parameters obtained from the training data using the population averaging (PA) method. From left to right, the plots display: Left, the optimal tolerance threshold across layers, Middle the number of neurons contributing to recurring embedding activity, and Right the maximal deviation threshold.

Note that the maximal deviation learned by the method typically increases from earlier to later layers. This trend reflects the architecture of LLaMA, where deeper layers exhibit a broader range of variability in their neural activity. In addition, the analysis reveals that concept encoding is generally supported by substantial populations of neurons rather than isolated units. Sparse encodings of concepts are rare. This suggest that concepts emerge through distributed patterns of activity across the network rather than through localized neural encoding.

### E.2 Sample Dictionary of Chunks Memorizing and Predicting Signals in Text

We then generalize this method to other locations of the token, such as prior to or following the occurrences of the signal at indices $V(s)$, denote the set of indices undergoing this $k$ step shift as $V^{-k}(s) = \{t - k \mid t \in V(s)\}, \quad V^{+k}(s) = \{t + k \mid t \in V(s)\}$. The former $(k > 0)$ examines the network's neural activity representing a memory of the signal, and the latter $(k < 0)$ examines the network's neural activity that is predictive of signals happening subsequent to the input prompt sequence.

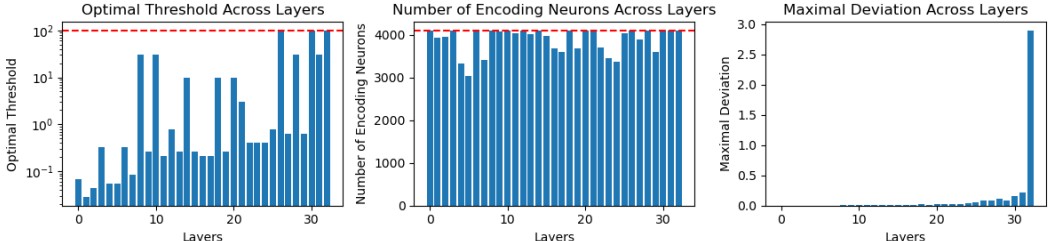

Figure 14: Layer-wise statistics of PA extracted chunks commonly preceding the concept "cake".

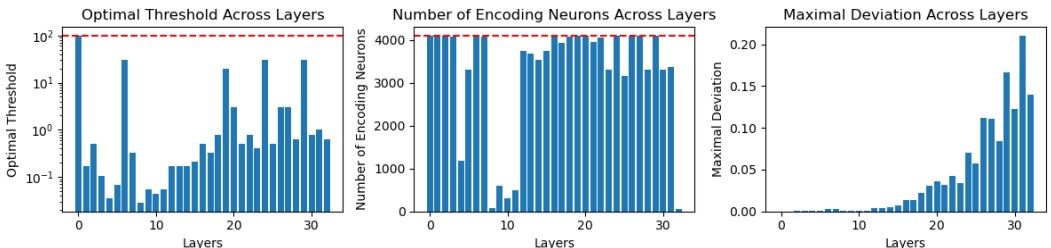

Figure 15: Layer-wise statistics of PA extracted chunks commonly preceding the concept "in".

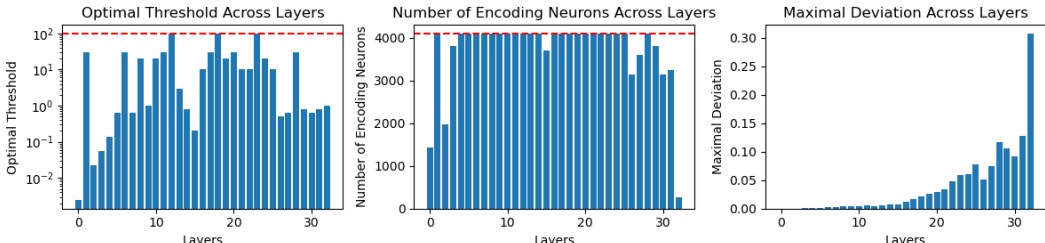

Figure 16: Layer-wise statistics of PA extracted chunks commonly proceeding the concept "of".

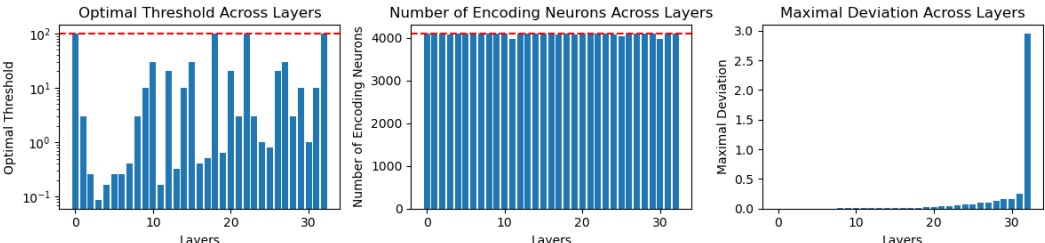

Figure 17: Layer-wise statistics of PA extracted chunks encoding the latest word being "people".

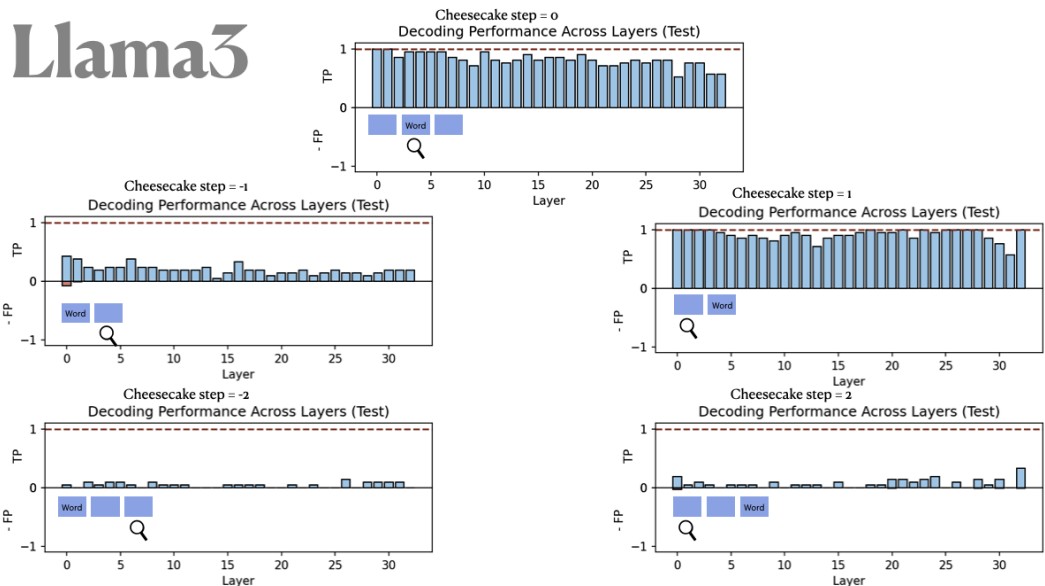

Figure 18: Evaluation of the decoding accuracy of the extracted chunks predictive of future elements (right panels) and memorization (left panels) of past elements beyond immediate next-token predictions (middle).

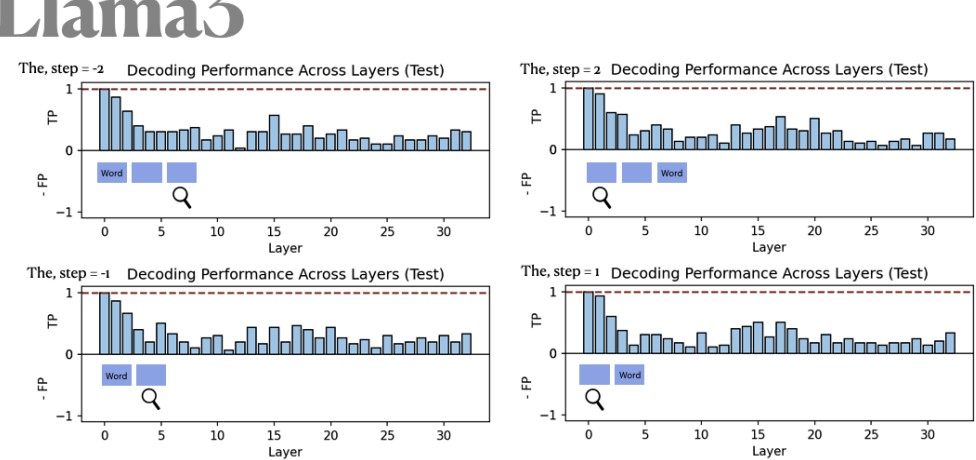

Figure 19: Another example evaluation of the extracted chunks predictive of future elements (right panels) and memorization (left panels) of past elements.

### E.3    Visualizing Concept Encoding Chunks at Different Sequence Token Positions

Figure 20 visualizes the subpopulation activity chunks extracted by the population-averaging method responding to the word "cheese", "cake", and "cheesecake" with their variety of tokenized form, ending with the current, previous, and subsequent signal indices. One observation is that the information about context is represented in the latter rather than the earlier layer of the network. Meanwhile, information about the memory of a signal was represented much more sparsely among the neurons in the network than the most recent signal. Additionally, the temporal coding of the signal is not uniform, representing the same word 'cheese' as memory, as the latest token, or as prediction is manifested in distinct neural population activities. There is a smaller neural population responsible for predicting a future signal than encoding for the current signal or the past signal. Evaluation of the decoding performance using the extracted chunks is shown in Figure E.3 and E.3.

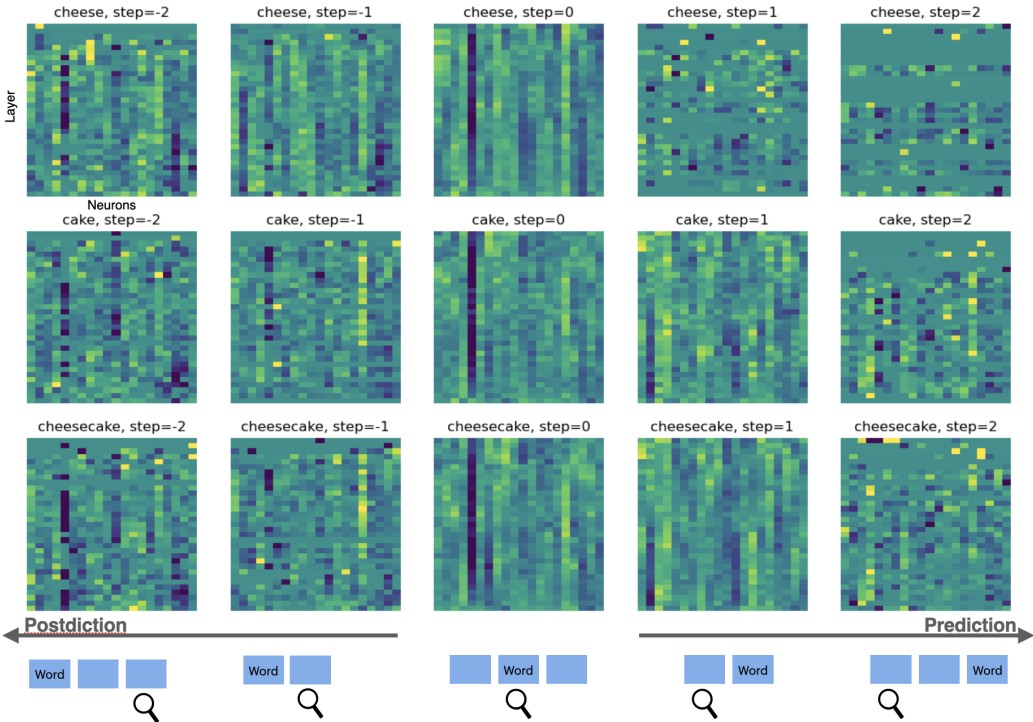

Figure 20: Visualization of the extracted neural subpopulation for the words "cheese", "cake", and "cheesecake", respectively, in both directions.

### E.4 Generalizing to Other Concepts

We used the population averaging method to extract neural subpopulation activity chunks of llama accounting for the 100 most frequently occurring words in English (as can be seen in main text Figure 5), adapting the threshold parameters on hidden activities collected from a training prompt and evaluating hidden activities collected from a test prompt (appendix H). For each word, we study subpopulation activities ranging from the two steps prior to and two steps subsequent to the last sequence token position. We observed subpopulation chunks that represent information in both directions.

We also show more examples of decoding accuracy to evaluate how indicative the extracted subpopulation chunk is of the signal's existence in Figure 21, 22, 23. Note that decoding performance in the 0th layer is usually perfect, as the 0th layer is the embedding layer having a fixed dictionary mapping tokens to the token embeddings. In subsequent layers, the presence of the extract indicates the occurrence of the recurring tokens in the input sequence. Having the recurring token in the input at the last sequence location elicits the network's hidden activity to lie within the range of the extracted embedding chunk.

Generally, the population averaging method learns chunks that are more predictive of the signal in the input sequence with words that have specific meanings than words that serve as prepositions; this can be caused by the network creating many neural population states to distinguish a preposition in different contexts, similar to the observation with RNNs (population averaging would be fragile in this case). We included more examples of chunk evaluation of this type.

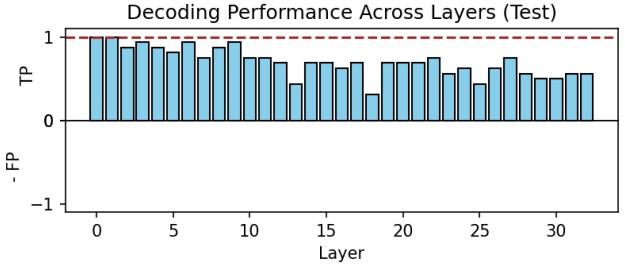

(a) word = people, step=-2

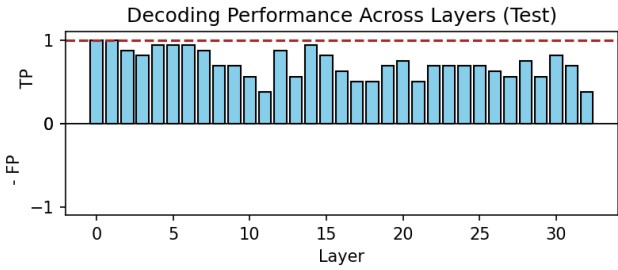

(b) word = people, step=-1

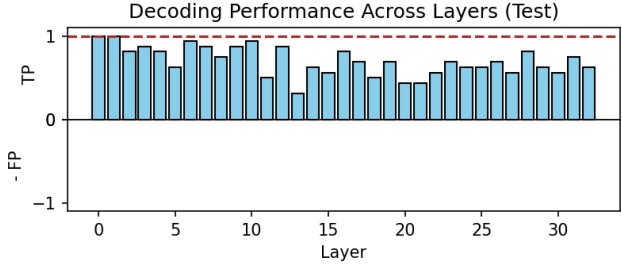

(c) word = people, step=0

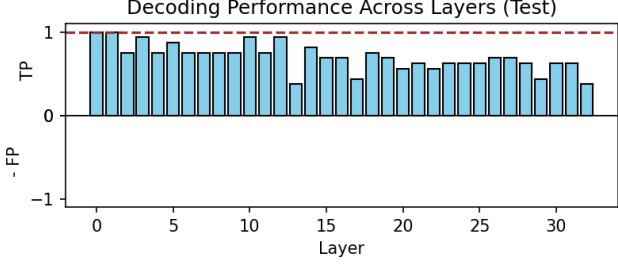

(d) word = people, step=1

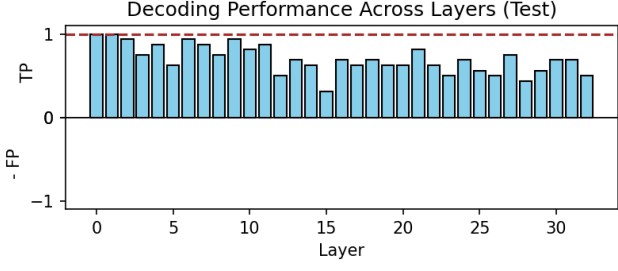

(e) word = people, step=2

Figure 21

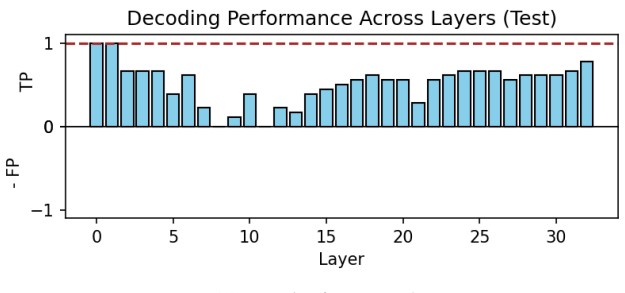

(a) word = it, step=-2

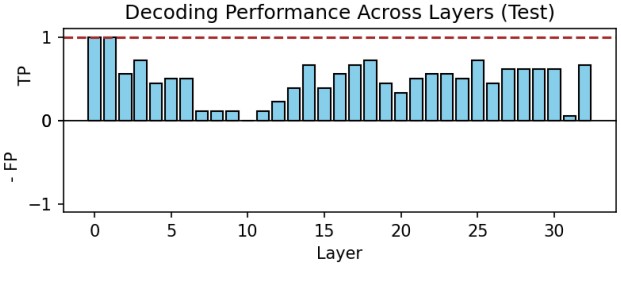

(b) word = it, step=-1

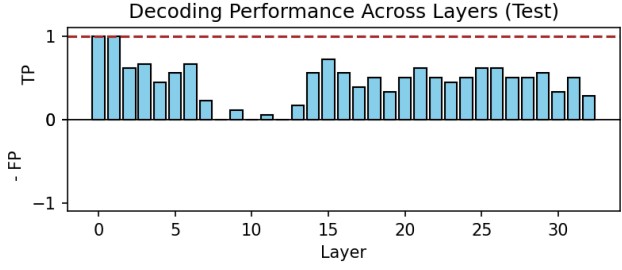

(c) word = it, step=0

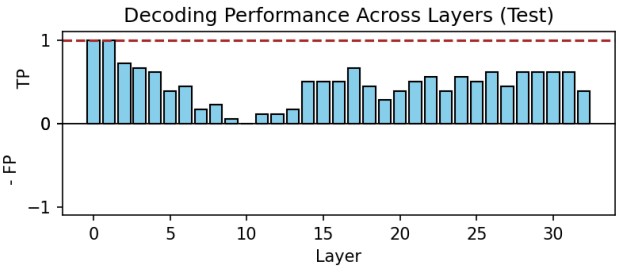

(d) word = it, step=1

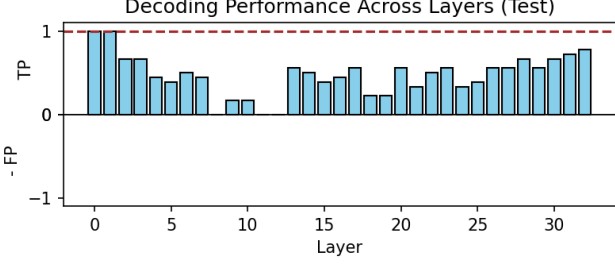

(e) word = it, step=2

Figure 22

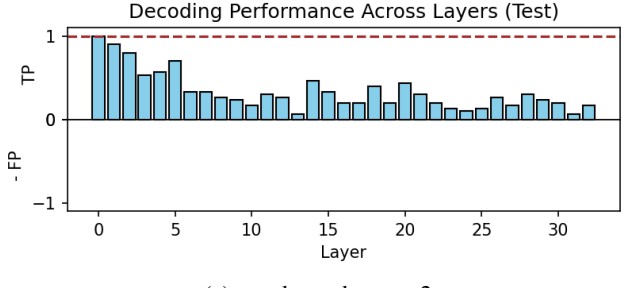

(a) word = and, step=-2

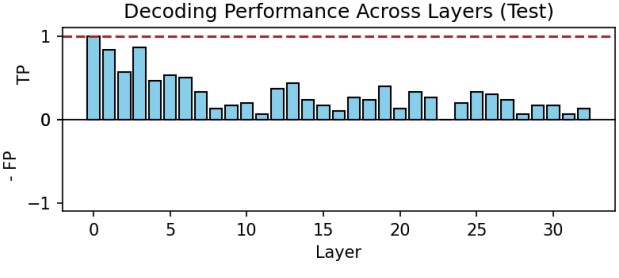

(b) word = and, step=-1

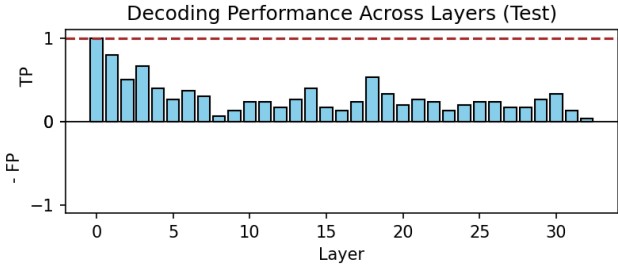

(c) word = and, step=0

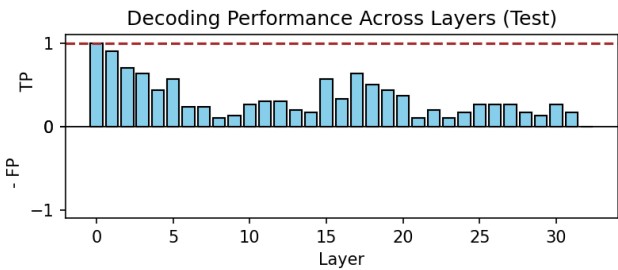

(d) word = and, step=1

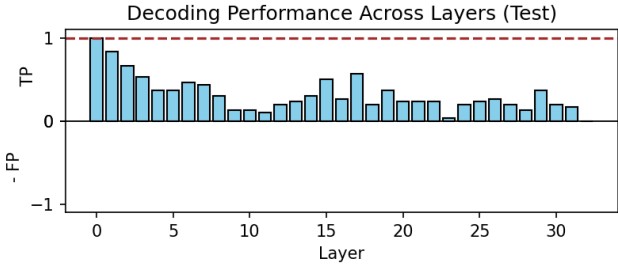

(e) word = and,step=2

Figure 23

### E.5 Extracting Chunks from a Range of LLMs

We here present more details of results evaluating the extracted PA chunks on other large models. Figure 24, 25, 26 shows the PA chunks corresponding to the sample concepts extracted from mamba (large-scale state-space model), T5 (encoder-decoder architecture), and RWKV (large scale RNN). From the left to the right are the decoding accuracy (divided into true positives and false positives) on the neural activities recorded from the test prompt; the optimal threshold extracted by PA across the layers; the number of concept-encoding neurons across layers; and the maximal deviation from the mean chunk prototype across layers.

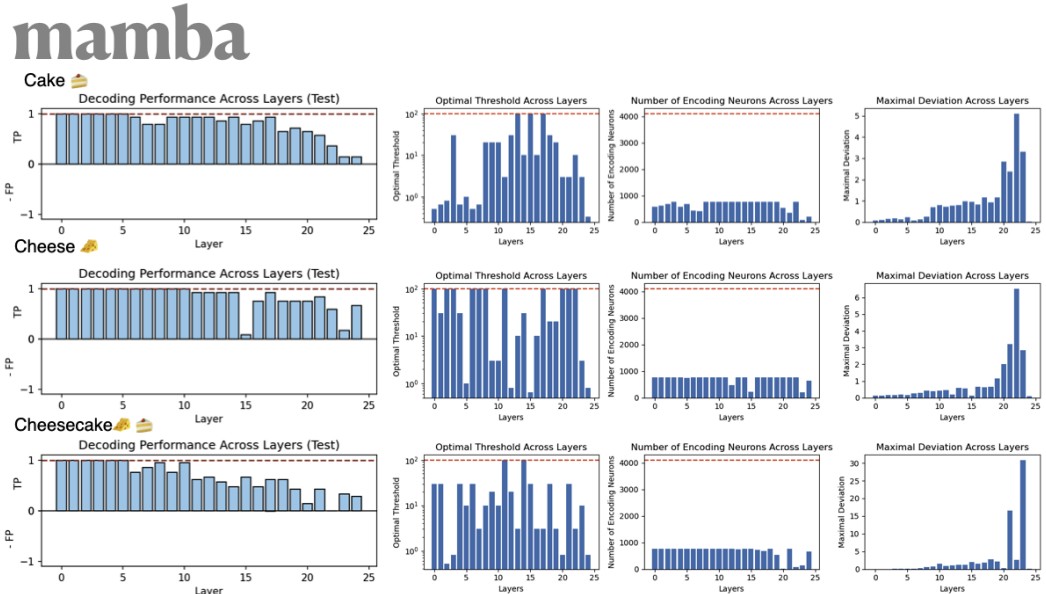

Figure 24: Evaluation of the extracted chunks in mamba (25 layers, embedding dimension = 768, large-scale RNN) on the three example words illustrated in the main paper.

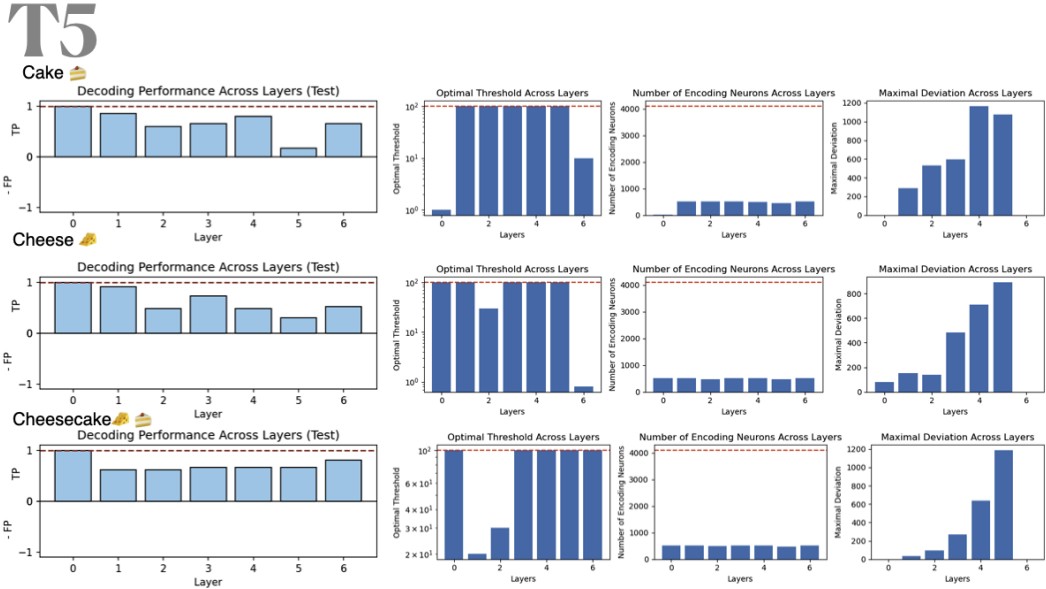

Figure 25: Evaluation of the extracted chunks in T5 (7 encoder layers, embedding dimension = 512, encoder-decoder architecture) on the three example words illustrated in the main paper.

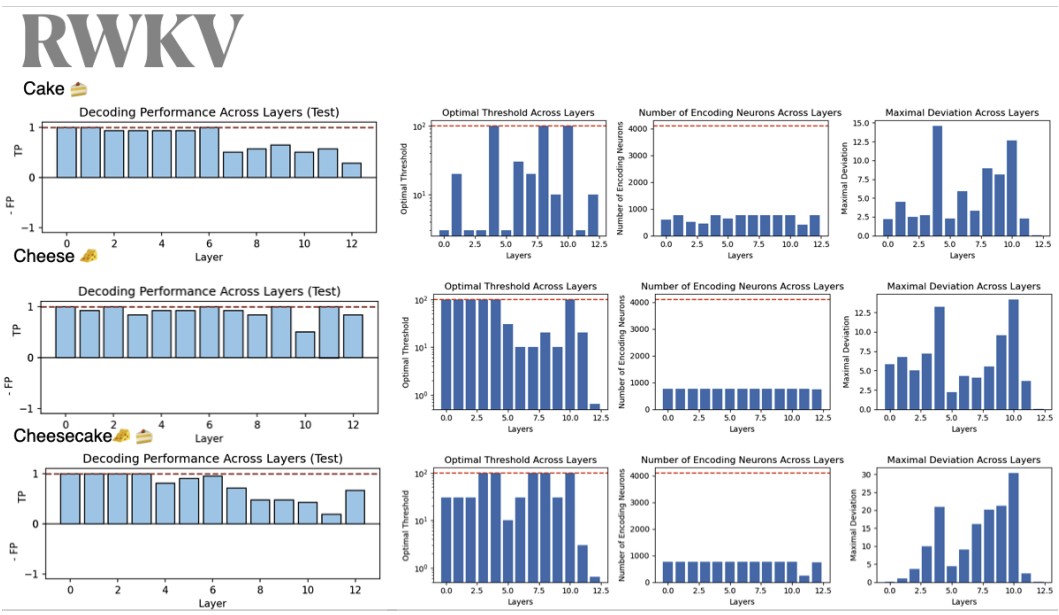

Figure 26: Evaluation of the extracted chunks in RWKV (13 layers, large-scale RNN) on the three example words illustrated in the main paper.

## E.6 Quantification on Causal Intervention

We quantified the effectiveness of causal evaluation by the percentage of concept occurrence in the generated sequences by LLM, comparing intervention (grafting/freezing) with control (no intervention) for the illustrated example concepts.

Table 3: Percentage of target concept occurrence for grafting and control using the prompt: "Hi, how are you doing?" ($N = 100$).

| Target Concept | Without Grafting | With Grafting |
|---|---|---|
| cake | 1% | 83% |
| cheese | 0% | 90% |
| cheesecake | 0% | 16% |

Table 4: Percentage of target concept occurrence for freezing and control across two prompts ($N = 100$). Row 1: Freezing results for the prompt: "What is the name of a rich, savory food made from curdled milk, often aged to enhance its flavors?" Target word: *cheese*, Perturbation Layers: 2–10, $N = 100$. Row 2: Freezing results for the prompt: "Name three most common desserts in Europe." Target concept: *cake*, $N = 100$.

| Target Concept | No Freezing | Freezing |
|---|---|---|
| cheese | 67% | 33% |
| cake | 36% | 9% |

Generally, grafting is more effective than freezing in influencing target concept occurrence. However, both types of intervention have measurable effects on influencing the model's generated text.

## E.7 Comment on Context Sensitivity of PA Chunks

Population averaging assumes that averaging hidden states across multiple contexts yields a meaningful, general 'chunk' for the concept. We study context-sensitivity of PA in extracting the internal representations in transformers. Indeed, modern language models such as Transformers are known to exhibit highly context-sensitive internal representations, and a given word (e.g., "bank") may evoke distinct activations depending on its context (e.g. a river bank versus a financial bank).

However, our goal is not to model context-specific usage, but rather to extract the core structural regularities that persist across contexts—analogous to approaching semantic prototypes in word2vec and GloVe. We assume that the mean of many contextual embeddings approximates a central semantic prototype, and averaging over many diverse instances of the identical concepts serves to extract the common representation shared across contexts.

This assumption on the algorithm then finds its empirical validation in the experiments. We observe that PA and UCD-derived chunks correlate well with many concepts. This suggests that for at least a subset of words, there exists a stable representation despite contextual variations. And this is the first step towards interpretability at a population dynamic level.

We acknowledge that this would be the case for some concepts, but sometimes multiple concepts map to the same word, such as 'bank' contains a different meaning on different occasions. In these cases there may be two chunks of neural population activities encoding for river bank and financial bank. And for some words such as prepositions, they have much more contextual nuance, we would expect this method to be less effective.

| Word | With Perturbation | Control (No Perturb.) | Δ (Perturb - Control) |
|------|-------------------|-----------------------|-----------------------|
| school | 78.7% | 0.7% | 78.0% |
| christmas | 74.3% | 0.3% | 74.0% |
| she | 71.7% | 0.0% | 71.7% |
| family | 71.3% | 1.7% | 69.7% |

Table 5: Words with the highest perturbation effectiveness (percentages).

| Word | With Perturbation | Control (No Perturb.) | Δ (Perturb - Control) |
|------|-------------------|-----------------------|-----------------------|
| if | 7.3% | 6.7% | 0.7% |
| at | 11.7% | 7.7% | 4% |
| on | 28.7% | 23.0% | 5.7% |

Table 6: Words with the lowest perturbation effectiveness (population averaging chunks).

Indeed we observed that the grafting success of the PA chunks is dependent on the type of words. We noticed grafting can be very successful on concrete words, examples the concepts most effective by grafting include school, christmas, family (table 5). For non-specific words such as prepositions and conjunctions, grafting can be ineffective. Examples are 'if', 'at', 'on' (table 6). This is often correlated with PA quality which can also be reflected in the PA decoding evaluation metrics for those specific concepts. In short, despite contextual dependency, empirical evidence suggests that this assumption works well for at least a substantial subset of all existing concepts, and this paves a way for future investigation to expand the interpretability step-wise towards more contextual subtlety.

## F    Unsupervised Chunking

### F.1    Attempted Architectures that Led to the Current Design

We also experimented with SAE-like variants for chunk discovery, using a reconstruction loss with continuous hidden activations and tying the decoder weights to the transpose of the encoder weights. However, this approach yielded too many hidden units to interpret meaningfully, and the continuous, additive nature of the reconstruction did not align well with our definition of chunks. To improve interpretability, we also tried variations that binarized the hidden layer, but this variant struggled to train effectively. Moreover, even with binarized activations, the additive reconstruction process still diverged from our intended definition of neural chunks as discrete, reusable population patterns.

### F.2    Evaluating Chunk Qualities

To evaluate the chunks extracted from UCD, we have explored a number of measures to assess the quality of the extracted chunks.

A good chunk needs to approximate the embedding. Therefore we check the normalized cosine similarity between the maximally matching chunk in the dictionary with each data point, and ensure

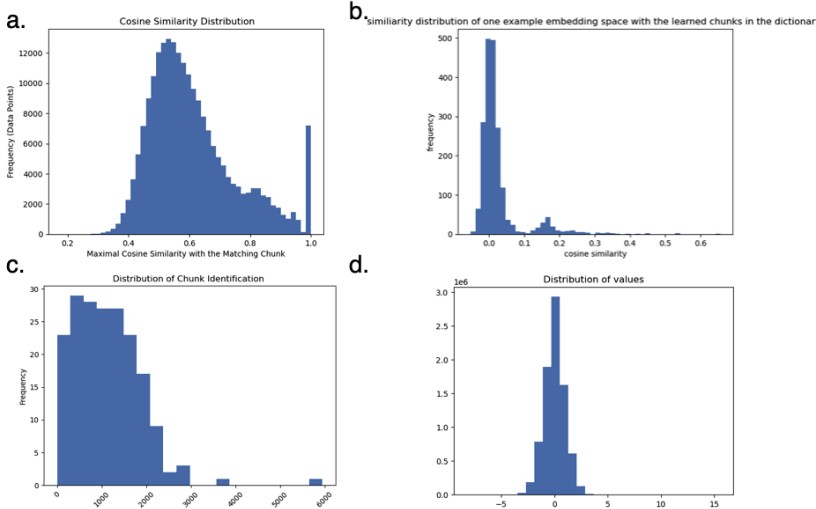

Figure 27: a. Similarity scores between the maximally similar chunk and the embedding it explains are largely unimodal, with a peak near one due to predictable tokens (e.g., start/end markers). b. Most embeddings are explained by a single dominant chunk, with low similarity to other chunks. c. Chunk identification frequency is unimodal; chunks are identified a similar number of times. d. Learned chunks lie in a sparse subspace, with weight distributions centered near zero.

that the cosine similarity between embedding and its matching kernel are reasonably similar. Figure 27 a shows the distribution of cosine similarities between embedding chunks and the embeddings in the 10th layer. The average cosine similarity between the maximally similar dictionary chunk and the embeddings is concentrated around 0.5.

To verify semantic distinctness, for one embedding, it needs to be similar to a few chunks while being different from most chunks in the dictionary. Figure 27 b shows the distribution of cosine similarities between embedding chunks and the embeddings in the 10th layer.

Additionally, the number of times that each chunk is identified to be the most similar chunk to the current embedding should be distributed in a reasonable way. There should not be a few chunks that are identified. This identification diversity is illustrated in Figure 27 c. Chunk usage is also unimodal, with most chunks selected a similar number of times.

Chunks lie in a sparse subspace with weights centered near zero (Figure 27d), and each embedding is typically explained by a single dominant chunk (Figure 27b). Figure 29 also shows two example chunk representations and the embeddings they explain, compared to control embeddings.

On top of that, we visually inspect samples of the extracted chunks, two of its matching embedding, and a non-matching embedding as a control. Two sets of these examples are shown in Figure 28.

### F.3  Visualization

Figure 28 a illustrates the most frequently identified chunk in the 10th layer (visualized as a $\sqrt{d} \times \sqrt{d}$ image, where $d$ is the embedding dimensionality), alongside two embedding examples where this chunk is identified as the maximally similar. Additionally, we include a control embedding where the same chunk is not identified as maximally similar. The extracted chunk is visually more similar to the two embedding examples than to the control embedding, demonstrating that this method extracts visually similar recurring neural population activities.

### F.4  The influence of hyperparameter K on Unsupervised Chunk Discovery

We tested the impact of varying K (500–8000) on early, middle, and late layers shown in 30. Increasing K does not consistently reduce loss, and the method's performance does not dramatically differ; overall, this method's effectiveness remains stable across a range of hyperparameter choices.

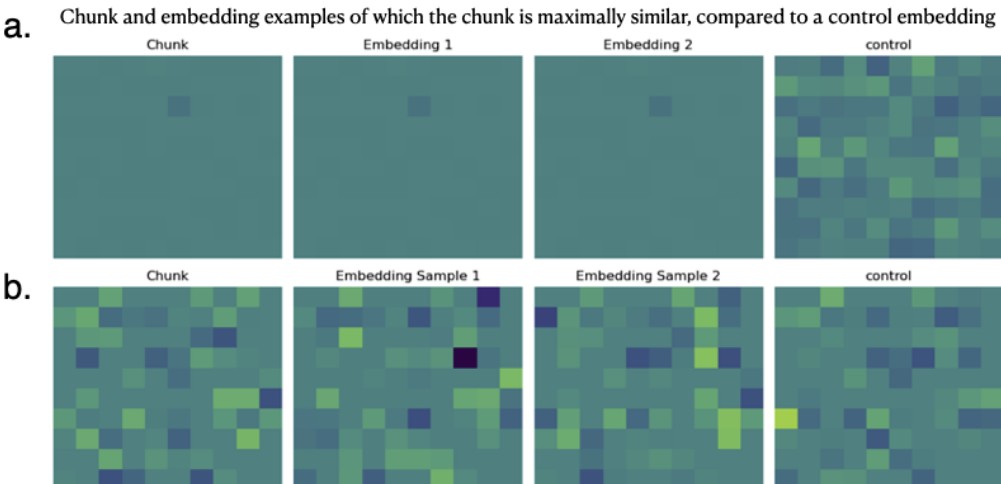

Figure 28: Two example chunk representations and the embeddings they explain, compared to randomly selected control embeddings. Chunk-associated embeddings exhibit high similarity and structure, while control embeddings show low similarity.

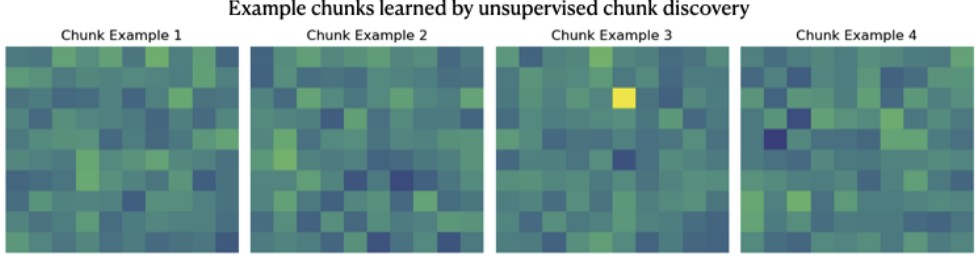

Figure 29: Example chunks learned by the unsupervised chunk discovery method.

### F.5  Full plot on layer-wise chunk processing in LLaMA-3

UCD simplifies high-dimensional activations into interpretable patterns of chunk appearance and disappearance. Figure 31 illustrates chunk interactions across LLaMA-3's all layers processing the beginning of *Emma*.

### F.6  Correlating sentence POS tags with chunk activities

We can interpret the unsupervised learned chunks by mapping them to interpretable linguistic structures, such as part-of-speech (POS) tags. To achieve this, we extracted the POS tags for the corpus using the averaged perceptron tagger [11], following the Penn Treebank POS Tagset [55]. We then computed the correlation between each chunk with each POS tag for every layer.

Figure 32c visualizes the maximum correlation between each POS tag and its most correlated discovered chunk across network layers (excluding the embedding layer). Our findings align with prior research showing that certain POS tags are processed in the earlier layers of the network [42, 82] and clearly demonstrate that there are chunk activities purely responsible for certain pos tags (possessive nouns, for example), while others hold very strong correlations.

Additionally, the POS-tag-correlated neural activities peak in earlier layers but also persist in later layers, indicating a sustained representation of syntactic information throughout the network. These findings suggest that chunk activities learned without supervision can serve as candidates for interpreting computational components within the network that are responsible for processing abstract concepts.

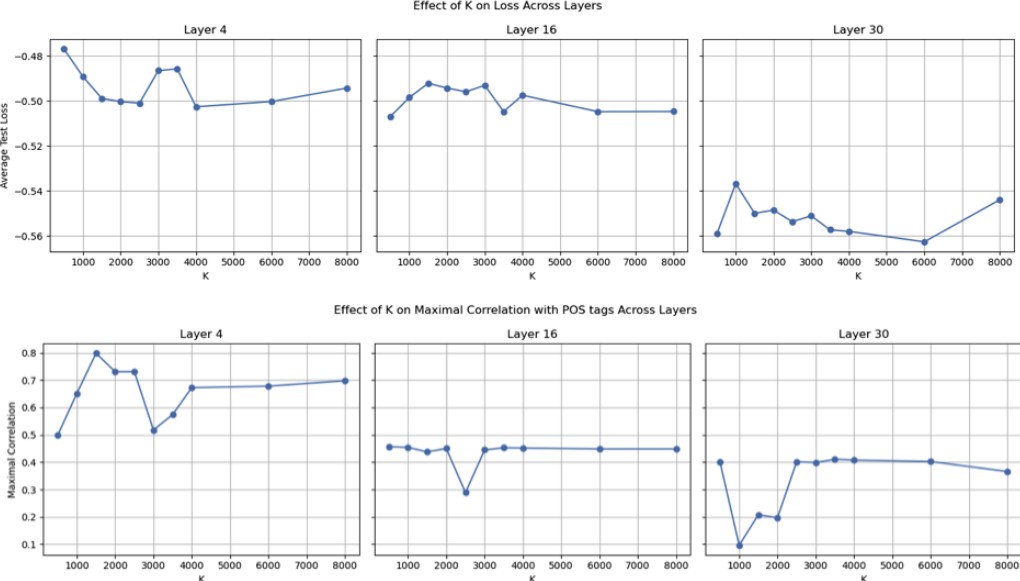

Figure 30: Effect of hyperparameter choice on the unsupervised chunk discovery method. The effect of K on the loss (upper plots) and maximal correlation with POS tags (lower plots) across layers 4, 16, and 30 (early, middle, and late). Increasing K does not consistently reduce loss, and the method's performance do not dramatically differ across choices of hyperparameters.

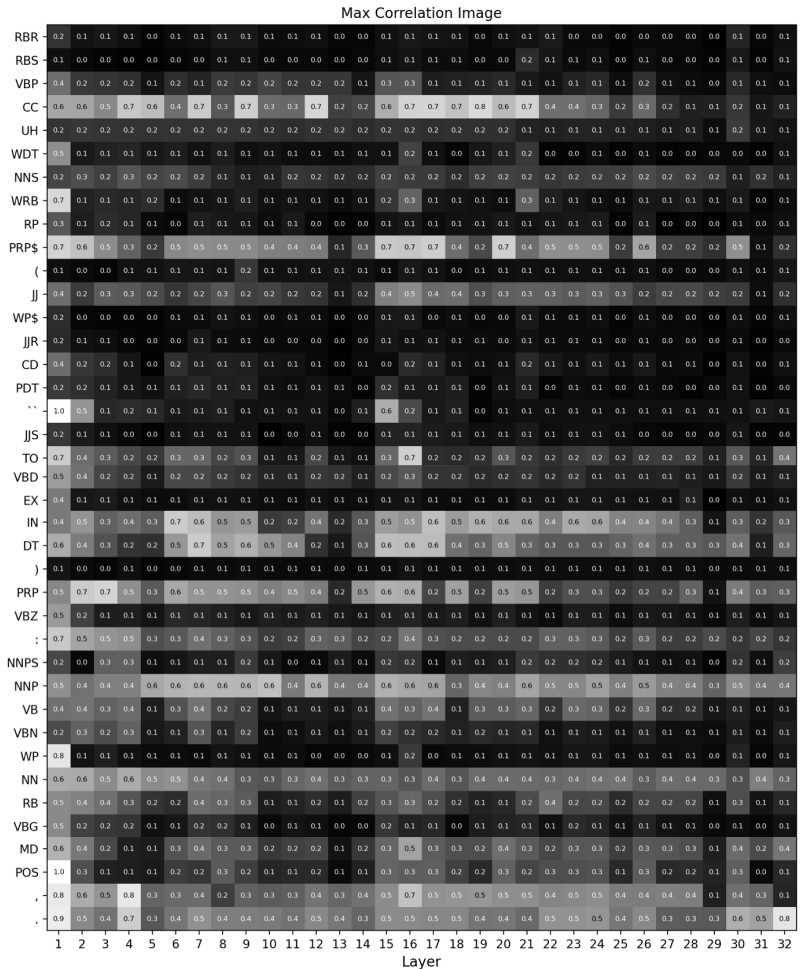

Figure 32: Part-of-speech tags and their maximal correlation across all layers in LLaMA-3.

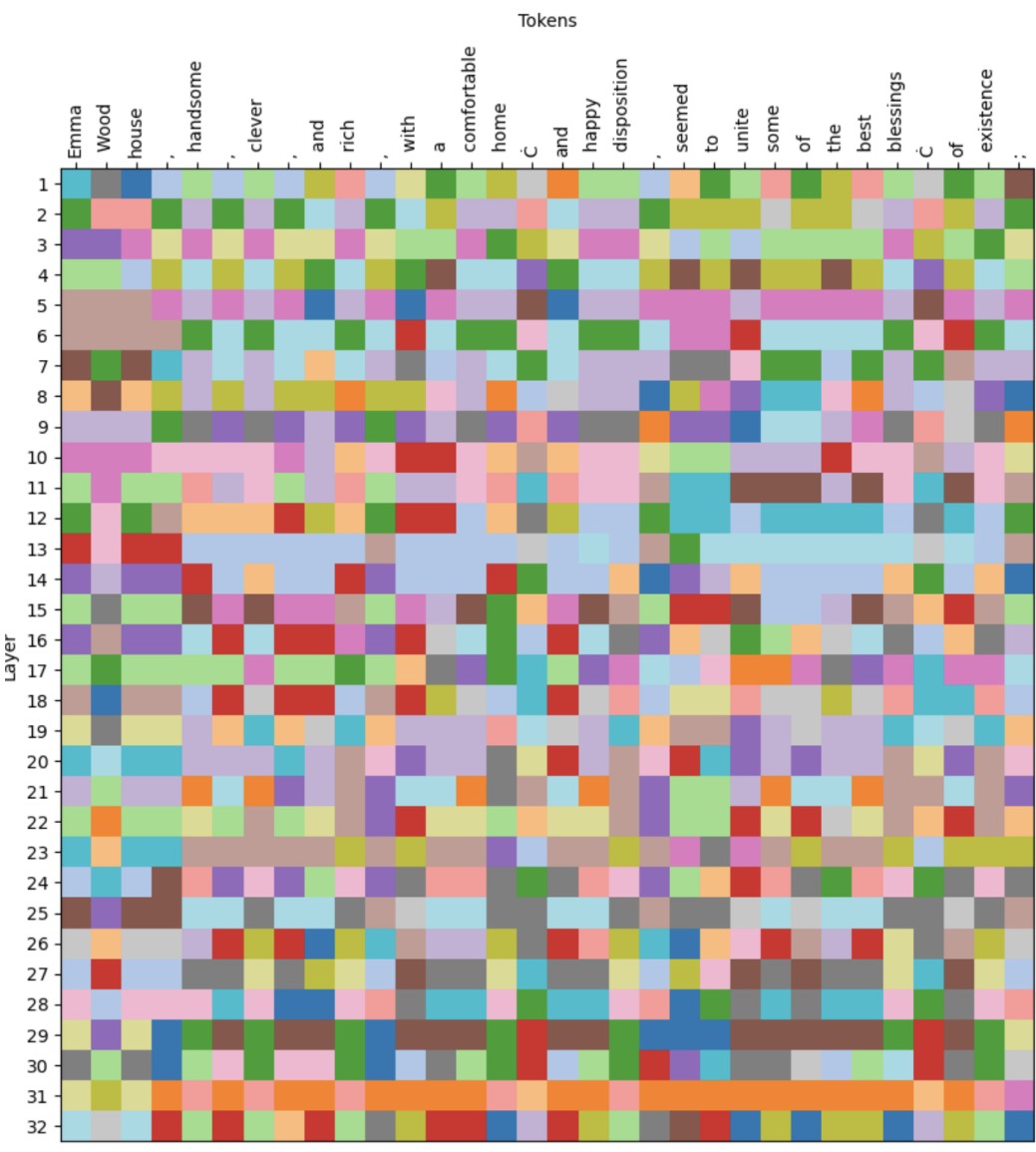

Figure 31: Interactions of neural population chunks upon parsing the beginning sentence of *Emma* (all layers).

# G    Computational Cost of Applying Chunk Extraction at Scale

All interpretability methods that analyze neural population activities need access to LLM neural activities, which can be acquired in a single forward pass on the data. This is required for SAE, representation engineering, and chunking methods alike.

For PA that extracts chunks from shorter input, taking a 1,000-token sequence, for example, our method is lightweight. The cosine-similarity-based matching and tolerance sweep are applied to hidden state arrays of ~16MB per layer, resulting in total memory usage under 200MB. Even across all 32 layers, the complete decoding procedure finishes in under 1 minute on a standard CPU. This makes our analysis tractable even in constrained environments.

The unsupervised chunk learning model consists of ~8.2 million parameters (learning a matrix of shape [2000, 4096]) and operates primarily through matrix multiplication for cosine similarity computation. With input sequences comprising ~200,000 tokens (e.g., Austen–Emma), the batch-based training regime (batch size 32) requires under 100MB of GPU memory per step. On a single NVIDIA RTX 3090, training for 1 epoch takes 30-60 seconds, and 100 epochs takes about 1-2 hours. All layers will take 32-64 hours if done sequentially. In comparison to the SAE variate, where the latent layer has 131,072 neurons (i.e., 32× the input dimension of 4096). This increases the parameter count to ~1.07 billion. The same setting completes 100 epochs in approximately 6–10 hours per layer. All layers will take 192-320 GPU-hours.

### Envisioning Chunk Discovery Scaling to Production-Sized LLMs (e.g., 70B+)

When applied to a production-scale model like LLaMA-70B, the proposed methodology remains tractable. For PA over sequences such as one containing 1000 tokens, each hidden state array occupies ~32MB, and the total memory usage across all 80 layers is approximately 2.5GB if done simultaneously. The total runtime will be a few minutes on CPU per concept across all layers. The unsupervised chunk learning model, which learns a $2000 \times 8192$ matrix (~16.4 million parameters), will require under 1GB of GPU memory per step and trains in approximately 2–3 minutes per epoch, totaling 3–5 hours for 100 epochs on a single NVIDIA RTX 3090, with batch size 32 and 200,000 input tokens. All 80 layers shall take 240 - 400 GPU-hours. In contrast, the sparse autoencoder variant with a 262,144-dimensional latent layer (32× overcomplete) increases the parameter count to ~4.1 billion, requiring ~50GB of memory and 12–20 hours to train over 100 epochs per layer. All 80 layers will take 860 - 1600 GPU-hours.

# H    Prompt Bank

We show the prompt used to extract the demonstrated word in the main manuscript using the population averaging method:

$prompt\_cheesecake\_train$ = "Cheese is one of the most versatile ingredients in the culinary world, and cheese can be used in everything from savory dishes to desserts. Cheese lovers often enjoy pairing cheese with crackers, wine, or fruit, but cheese also shines in baking. Cake, on the other hand, is the quintessential dessert, with cake being a staple at celebrations. Cake comes in many forms, such as chocolate cake, vanilla cake, or even carrot cake. However, when you bring cheese and cake together to create cheesecake, a magical transformation happens. Cheesecake is a dessert like no other, with cheesecake offering the creaminess of cheese and the sweetness of cake in perfect harmony. Cheesecake can be topped with fruits like strawberries or blueberries, or cheesecake can be flavored with chocolate or caramel. Some people prefer classic cheesecake, while others enjoy a more decadent cheesecake loaded with toppings. Regardless of the variation, cheesecake remains one of the most beloved desserts worldwide. The crust of cheesecake, often made from crushed biscuits or graham crackers, complements the smooth filling, making cheesecake irresistible. Cheese plays a central role in cheesecake, while the influence of cake ensures that cheesecake is always a delightful dessert. Whether you love cheese, crave cake, or are obsessed with cheesecake, this dessert proves that the combination of cheese and cake is truly extraordinary. Every bite of cheesecake reminds us that cheese and cake, when united in cheesecake, are a match made in heaven. Cheesecake aficionados often debate whether baked cheesecake or no-bake cheesecake is superior, but all agree that cheesecake is a dessert worth savoring. With so many variations, cheesecake enthusiasts never tire of exploring new ways to enjoy their favorite dessert. Cheese and cake come together seamlessly

in cheesecake, showing how cheese and cake can create something greater than their individual parts. Cheesecake is, without a doubt, the ultimate testament to the greatness of cheese and cake in unison."

The extracted chunks are then evaluated on the recorded hidden activity of the following prompt:

$prompt\_cheesecake\_test$ = 'Cheese is a culinary treasure that has delighted taste buds for centuries. Whether it's creamy, tangy, or sharp, cheese offers endless possibilities. Cheese finds its way into countless dishes, from savory casseroles to gooey pizzas, and its versatility knows no bounds. Cake, too, is a universal favorite, with cake symbolizing joy, celebration, and indulgence. Cake comes in every flavor imaginable—chocolate cake, vanilla cake, red velvet cake—and each cake brings its own special charm. But when cheese and cake are combined to form cheesecake, something truly extraordinary happens. Cheesecake is a dessert that transcends expectations, merging the velvety richness of cheese with the sweet, airy allure of cake. Cheesecake can be baked or chilled, simple or elaborate, yet every cheesecake captures the perfect balance of flavors. Classic cheesecake recipes highlight the creamy taste of cheese, while fruit-topped cheesecake adds a burst of freshness. Some people swear by chocolate cheesecake or caramel-drizzled cheesecake, while others can't resist a zesty lemon cheesecake. No matter the variation, cheesecake consistently proves that the union of cheese and cake is a match made in heaven. The crust of cheesecake, typically crafted from crushed cookies or graham crackers, provides the ideal foundation for the smooth and luscious cheese layer. Every bite of cheesecake reminds us why this dessert has stood the test of time. Fans of cheese, cake, and cheesecake alike agree that cheesecake combines the best of both worlds. Whether you're indulging in a slice of classic cheesecake, exploring new cheesecake flavors, or savoring the rich taste of a perfectly baked cheesecake, it's clear that cheese and cake reach their pinnacle when united in cheesecake. Cheesecake is a testament to how cheese and cake, when brought together, create something greater than the sum of their parts. From the first bite to the last, cheesecake is a celebration of everything wonderful about cheese, cake, and, of course, cheesecake itself.'

We show the prompt used to extract the top 100 frequent words using the population averaging method:

$prompt\_frequent\_words\_train$ = "In today's world, people often feel the push and pull of connection and solitude. With technology and social media on the rise, we now have countless ways to stay in touch with those we know and love. However, the question of whether this digital world can truly satisfy our need for real connection remains. To be truly connected is to share experiences, to understand and to be understood. This kind of connection goes far beyond a screen. The desire to connect is universal, and people have searched for it throughout history. In ancient times, communities formed to support one another, to live together, and to build bonds that could help them through challenges. Today, people may still crave this closeness, yet it is not always easy to find in our fast-paced world. While the internet gives us access to almost anyone, anywhere, it does not always give us the depth of connection that true friendship and family relationships can offer. Consider a family spending time together. For many people, family is a source of strength, a place where they feel safe and understood. But as life becomes busier, it can be easy to let work, hobbies, or other commitments pull us away from family time. Many people find that they must make a conscious effort to set aside time for their loved ones. To sit down for a meal together, to talk about the day, to share thoughts and laughter — these moments are priceless. They are what remind us of who we are and who we want to become. Friendship, too, plays a significant role in life. Friends are often the people we choose to spend time with, the ones who share our interests and support us. To have friends is to feel understood in a unique way, to laugh together, to encourage each other, and sometimes just to sit in silence knowing someone is there. A true friend listens without judgment, stands by us in hard times, and celebrates with us in good times. However, maintaining friendships can require work and commitment. With busy lives, people can sometimes lose touch, only to realize later how much those friendships meant. In the workplace, relationships are equally important. Many people spend a significant amount of time at work, so having good connections there can make a big difference. Working with others requires collaboration, understanding, and respect. When people feel connected to their colleagues, they tend to work better together, share ideas freely, and support each other. A positive work environment can foster not only productivity but also well-being. People are more likely to feel satisfied in their work when they know they are valued and understood by those around them. Of course, technology plays a large role in modern connections. Platforms like social media allow people to connect across distances, to share life events, and to communicate instantly. For some, these tools make it easier to stay in touch with family and friends, to share news, and to express themselves. However, while technology can bring people closer, it can also create a sense of

distance. Seeing the lives of others through a screen is different from experiencing life together in person. The highlights of life shared online may not always show the full picture, leaving people to wonder if they are missing out. To have genuine connection, people often need to go beyond what is easy and convenient. Sometimes, this means reaching out, making an effort, and being open. True connection requires understanding and empathy. It asks us to listen, to be present, and to care. In a world that often moves fast, taking the time to connect deeply can feel like a challenge, but it is also incredibly rewarding. The value of connection is evident in difficult times. When people go through challenges, it is often those close connections that help them through. Whether it's a friend who listens, a family member who offers support, or a colleague who steps in to lend a hand, these connections give people strength. Knowing that someone else understands or is there to help can make all the difference. Moreover, the ability to connect also fosters compassion. When people share experiences, they begin to see the world from each other's perspectives. This understanding can lead to greater kindness and less judgment. People who feel connected are often more empathetic, more understanding, and more willing to help others. This creates a positive cycle, as kindness and empathy tend to inspire more of the same. For people to have a balanced and fulfilled life, connection is essential. But to nurture these connections takes effort. It is not always easy to set aside time, to reach out, or to stay in touch. Life can be busy, and distractions are everywhere. However, those who make the effort to connect often find that their lives are richer and more satisfying. The joy of shared laughter, the comfort of understanding, and the strength of support are all things that make life meaningful. As we move through life, the connections we make help to shape who we are. We learn from others, grow with them, and find new perspectives. Each person we meet adds to our experience and helps us to see the world in new ways. Sometimes, people find that their most valuable lessons come from those who are different from them. To connect with people from various backgrounds and with different life experiences is to broaden our view of the world. In conclusion, to live fully is to connect meaningfully. Whether through family, friends, colleagues, or even strangers, these bonds enrich life. They offer joy, comfort, and understanding, and they remind us that we are not alone. While technology may change the way we communicate, it cannot replace the depth of real connection. To make time for those who matter, to share moments, and to care is to live a life of purpose and love."

The extracted chunks on the top 100 frequent words are then evaluated on the recorded hidden activity of the following prompt:

$prompt\_frequent\_words\_test$ = "In the world we live in, each day is filled with choices we all make, big or small. The way we approach these choices can be what shapes not only our own lives but also the lives of those around us. To be the kind of person who reflects on what they do, who they are, and who they want to become, is to take a meaningful step toward self-awareness and growth. One of the first things to know about making decisions is that they are all interconnected. When we choose to do one thing, it often means we cannot do another. This may seem obvious, but to understand the full impact of this reality, it helps to look at the ways in which choices play out in real life. We are always presented with options, and while some decisions may seem trivial at first, they add up over time. We can think about it like this: to choose a path, even if small, can set in motion a series of events that shape our lives in ways we could never fully predict. For example, people make decisions on how to spend their time. Time is one of the most valuable resources we have. There is no getting more of it, and once it's gone, it's gone for good. How we choose to spend it — whether working, relaxing, being with family, or helping others — says a lot about what we value. Some people may spend time worrying about things that, in the end, are not as important as they seem, while others may put time into making themselves or others better. This shows the difference in what people find to be meaningful or valuable in life. When we look at those around us, we see that everyone is trying to figure out what it means to live a good life. Some believe that to have a successful career is key, while others might think that family or friendships are the foundation of a fulfilling life. Whatever the focus may be, it's clear that we all want a life filled with purpose. The concept of purpose is one that everyone seems to look for, although it can mean different things to different people. People sometimes find that purpose comes from the roles they play in the lives of others. For instance, many parents feel that to have children and raise them well is one of the most meaningful things they can do. They look to guide their children, to give them the tools they need to make their own choices. The idea of helping others extends beyond family, as people also contribute to their communities in many ways. Volunteering, supporting friends, and giving back are just some ways people find meaning in their lives. In work, too, people seek purpose. It's common to find that people want to do something that they can be proud of, something that allows them to use their

talents and contribute to society. This desire to work well is what drives many people forward and keeps them motivated. Yet, work can also become overwhelming, especially when people forget to balance their time between work and other areas of life. Balance is essential in any good life. We often have to remind ourselves not to let any one part of life take up all of our attention. It's not easy, but it is essential if we want to live fully. This balance extends to how people think about success. For some, success is about achieving certain goals, like owning a home, getting a promotion, or earning a certain amount of money. For others, success is about having good relationships, feeling at peace, and being happy. Each person will have their own idea of what it means to succeed. Some may be quick to compare themselves to others, thinking that if someone else has something they don't, they are somehow lacking. But to make comparisons is not often helpful. We each have our own journey, and to look too much at what others have can take away from the joy of our own experiences. Another important part of life is facing challenges. There are times when things don't go as planned, and it's easy to feel frustrated. However, these moments are often when we learn the most about ourselves. Challenges can show us what we are capable of and remind us that we are stronger than we think. People often say that to know hardship is to know strength, and it's true that the challenges we face can make us wiser and more resilient. People also face choices about how they treat those around them. Kindness, empathy, and patience are qualities that many strive to have, but it's not always easy to be kind in a busy, fast-paced world. However, those who make a habit of treating others with respect and understanding often find that they are happier and more fulfilled. Relationships, whether with family, friends, or colleagues, require effort and care. By giving time to nurture these connections, we make life richer not only for ourselves but also for others. In moments of reflection, people often think about their lives and ask themselves, "Am I doing what I want to do? Am I living the life I want to live?" These questions can be difficult to answer, but they are important. To pause and look within is to take stock of what matters. This reflection can help guide future decisions and keep people on the path that is right for them. At the end of the day, life is made up of moments, decisions, and interactions. To be mindful of how we choose to spend these moments, of who we choose to spend them with, and of what we put into the world, is to live with purpose. While each person's path may be unique, we all share the desire to find happiness and meaning. In seeking to make each day count, to treat others well, and to pursue goals that matter to us, we contribute to a world that is better for all. It may not always be easy, and we may not always make the right choices. But to try, to reflect, and to grow from each experience, is to live a life well-lived. We each have the power to make choices that, in time, build a story of who we are and what we stand for. Let that story be one of kindness, purpose, and joy, shared with those who mean the most to us."

## I  Compute Disclosure

All experiments were conducted on a shared internal cluster equipped with NVIDIA Quadro RTX 6000. Collecting data on neural activities from LLaMA-3-8B residual streams took approximately 25 GPU hours, and from SAE neurons took 5 GPU hours. Training the unsupervised chunk discovery across 32 layers took about 90 GPU hours. Additional exploratory and ablation experiments, which were not included in the final version of the paper, consumed approximately another 50 GPU hours. The total compute used for the experiments reported in the paper is estimated at around 170 GPU hours. We include these details in the supplementary material to facilitate reproducibility.

## J  Ethics Disclosure and Broader Impact Statement

This work contributes to the interpretability of large-scale neural networks by identifying recurring structured patterns ("chunks") in their internal representations. On the positive side, enhancing the transparency of these models can aid in better debugging, auditing, and alignment with human expectations, potentially reducing harmful outputs and supporting trustworthy AI deployment in high-stakes settings such as healthcare, education, or legal decision-making.

Furthermore, our chunk-based approach can facilitate more sample-efficient or modular fine-tuning by enabling structured interventions in network behavior, which may reduce computational costs and environmental impact in downstream applications.

However, we also acknowledge that improved model interpretability may lower the barrier to misuse by enabling more precise model manipulation, adversarial targeting, or extraction of private information from model internals. Additionally, interpretability methods may be incorrectly assumed

to confer safety or fairness guarantees when, in reality, they offer only partial insights. These risks underscore the importance of using interpretability tools in a contextual and ethical manner, especially in deployment.

We encourage responsible application and further research into robust, human-aligned interpretability frameworks that include social considerations beyond technical performance.

