# OpenReview forum: "Concept-Guided Interpretability via Neural Chunking"
_NeurIPS.cc/2025/Conference — NeurIPS 2025 poster_

### Official Review · Reviewer_fcFE · 2025-06-30

**Clarity:** 2
**Significance:** 3
**Originality:** 3
**Rating:** 5
**Confidence:** 4

**Summary:**

The paper introduces a new framework for interpretability, inspired by cognitive science. The main claim is that neural networks’ activations mirror the statistical properties of the distributions they model, termed the Reflection Hypothesis. The authors propose several dimensionality reduction algorithms to extract interpretable concepts from the activations and show that the concepts causally impact NN behavior.

**Questions:**

1. Could there be more discussion and exploration of other possible unsupervised algorithms for chunking? I would consider raising my score if there were interesting, novel results there.

**Ethical Concerns:**

["NO or VERY MINOR ethics concerns only"]

**Final Justification:**

Added discussion about more methods which suggested promising results

**Limitations:**

no, please discuss possible limitations of when the reflection hypothesis might break down or when it might be difficult to extract insights from the activations.

**Quality:**

3

**Strengths And Weaknesses:**

## Strengths
- The paper introduces a novel framework for conducting future interpretability experiments that is grounded in cognitive science, rather than more mechanistic or representational approaches. The framework offers a human-in-the-loop approach to interpretability that may pair well with AI agents.
- The paper shows the hypothesis across multiple network architectures and that it can lead to interpretable concepts, such as narrative structures in passages. These appear to be novel insights that can drive NN behavior.

## Weaknesses
- Some of the results seem to be reformulations of previously known results in interpretability, e.g., that "Recurring chunk patterns emerge, such as similar activations for commas and adjectives in early layers" is similar to the findings in [1]
- Two of the proposed methods seem somewhat impractical. Discrete sequence chunking is mostly tenable when the NN’s hidden dimension $d$ is somewhat small. The authors then propose Neural Population Averaging, but this requires some hypothesis as to the concepts learned by the network. Specifically, the authors write that “this method relies on prior knowledge of recurring patterns in the input sequence”. Thus the main practical approach is unsupervised chunk discovery, but the only algorithm the authors give here is a variant of a k-means clustering algorithm.

[1] https://karpathy.github.io/2015/05/21/rnn-effectiveness/

---

> ### Author Rebuttal · Authors · 2025-07-30
>
> We thank the reviewer for the comments. We are reassured that the reviewer found the proposed framework novel. We address the rest of the comments below:
>
> > [W1] Some of the results seem to be reformulations of previously known results in interpretability, e.g., that "Recurring chunk patterns emerge, such as similar activations for commas and adjectives in early layers" is similar to the findings in [1]
>
> Thanks for suggesting related studies. In this blog, Karpathy trained RNNs to generate character-level sequences such as Shakespearean text and Linux source code, and identified individual neurons that responded to features like quotation marks, if statements, or line endings. However, he also noted that the majority of neurons lacked clear interpretability, raising the question of how to understand model behavior at the population level when single-neuron activations are ambiguous. This early investigation inspired many subsequent studies focused on interpreting individual neurons, but progress stalled as it became evident that most neurons are polysemantic (Elhage et al., 2022) and do not correspond to unique, human-interpretable concepts.
>
> In contrast to Karpathy’s individual neuron approach, our work suggests that concepts are not encoded by the activations of individual neurons, but rather by subpopulations of neurons whose joint activity visits similar regions in the high-dimensional population space. Instead of relying on neuron-wise interpretation, we propose methods to analyze and localize concept-encoding subpopulations—advancing toward interpretable representations at the population dynamics level.
>
> > [W2] Two of the proposed methods seem somewhat impractical. Discrete sequence chunking is mostly tenable when the NN’s hidden dimension is somewhat small. The authors then propose Neural Population Averaging, but this requires some hypothesis as to the concepts learned by the network. Specifically, the authors write that “this method relies on prior knowledge of recurring patterns in the input sequence”. Thus the main practical approach is unsupervised chunk discovery, but the only algorithm the authors give here is a variant of a k-means clustering algorithm.
>
> We thank the reviewer for raising concerns about the practicality of our methods. We respectfully disagree with the notion that the approaches are impractical and would like to clarify several points.
>
> While Neural Population Averaging (PA) indeed relies on recurring patterns in the input (as we state), this is actually a strength, not a limitation. In our examples we also simply searched for the concepts by pattern matching using regular expressions. In many data domains practitioners already know the types of concept they are looking for, and abundant data are labelled by their keywords, abstract schema, intent, and there are tools to extract abstract structures such as pos tags or sentence parsing trees. In a range of corpora, PA allows us to isolate and interpret these reusable representations without retraining the model. Our grafting experiments show that these prototypes generalize systematically (see discussion with reviewer 1jWN for details).
> We will expand our supplementary material to provide more technical detail on our implementation and clarify how pragmatic the proposed methods are. (Also see discussion with reviewer 86W2 for compute estimate)
>
> > [Q1] Could there be more discussion and exploration of other possible unsupervised algorithms for chunking? I would consider raising my score if there were interesting, novel results there.
>
> Yes, we have explored a couple of alternatives.
>
> We also experimented with SAE-like variants for chunk discovery, using a reconstruction loss with continuous hidden activations and tying the decoder weights to the transpose of the encoder weights. However, this approach yielded too many hidden units to interpret meaningfully, and the continuous, additive nature of the reconstruction did not align well with our definition of chunks. To improve interpretability, we attempted to binarize the hidden layer, but this variant struggled to train effectively. Moreover, even with binarized activations, the additive reconstruction process still diverged from our intended definition of neural chunks as discrete, reusable population patterns.
>
> We explored several variations of the unsupervised chunk discovery (UCD) loss. One variation is maximal cosine similarity + sparsity regularization, which yielded to us an average of 0.54 cosine similarity from the dictionary. We also tried instead of the max, a softmax weighted similarity with a sparsity regularization, and that yielded to us the best case of 0.58 cosine similarity distribution in the test data. Removing the sparsity regularization and simply using the max gives us 0.60 average cosine similarity. For each configuration, we also evaluated the quality of the chunks by semantic distinctness, identification frequency, identification diversity (illustrated in Figure 27 in Supp.), and visualized and inspected them (Figure 28). The additional component did not contribute to learning a better dictionary, so we decided to report the simplest and the most effective method. We also studied whether the size of the dictionary matters and did not observe a significant effect of K (Supp. E.5 Figure 32).
>
> But UCD reveals very interesting computational structures in LLMs that call for a closer look in future investigations. Such as the connectivity between chunks across layers: chunk 412 in layer 1 elicits chunk 1621 or chunk 1251 in layer 2 (L1:412 → L2:1621) (L1:412 → L2:1251), but not other chunks in layer 2. This allows the extraction of a graph of neural chunk interaction across all 32 layers of llama. We found that some layers act as hub layers, such as layer 31, where certain chunks receive input from many chunks in the preceding layer, aggregating diverse upstream activity. In contrast, other layers—like layer 30—function as expansion layers, generating a large number of distinct chunks to differentiate activity patterns inherited from the previous layer. Interestingly, we also observed a persistent chain of chunk activations spanning from the first layer to the final layer of LLaMA. This suggests that a specific chunk—likely originating from the input and activated in the first layer—continues to activate a unique chain of chunks across all 32 layers. This behavior may reflect a positional marker in the sequence, such as a beginning-of-sentence or end-of-sentence token. Such a chunk interaction graph gives a discrete description over the neural population interaction mechanisms inside the large models, and can be a subject under investigation and perturbation in diverse usage cases.
>
> Besides that, UCD reveals ample amounts of temporal patterns of chunk interactions, such as in the Emma corpus, chunk 1145 136-1300-367 occurs 80 times, 1145-1300-367-367-367 occurs 115 times. To quantify these extracted patterns of temporal computation dynamics, we computed a compression ratio—the number of input tokens divided by the number of extracted population-level patterns covering the same corpus for each layer.
>
> | Layer | Compression Ratio |
> |-------|--------------------|
> |   5   | 0.39               |
> |  10   | 0.45               |
> |  15   | 0.62               |
> |  20   | 0.55               |
> |  25   | 0.43               |
> |  30   | 0.63               |
>
> This suggests that neural activity exhibits structure not only at the token level but also across multiple tokens, indicating the presence of higher-order, compressible computation dynamics.
>
> While we are still actively developing improved versions of this algorithm, we include our current best iteration in this paper. We believe this method offers a promising starting point for identifying relational entities from neural activity and provides a foundation for future work aimed at uncovering the rules governing neural population interactions in task-specific computations.
>
> > [L] please discuss possible limitations of when the reflection hypothesis might break down or when it might be difficult to extract insights from the activations.
>
> The reflection hypothesis may fail for networks that are insufficiently trained. This is evident in the RNN experiment with sequences such as ABCD and CDAB. When undertrained, the network does not differentiate between the shared subsequence CD in the two contexts. Instead, it learns a shallow prediction pattern for AB and CD, a tendency that is also observable in its neural trajectories. However, in these two cases, CD should lead to different predictions: in ABCD, it precedes E, whereas in CDAB, it is followed deterministically by A. The network gradually acquires this contextual sensitivity over the course of training. Similarly, we observe that when we make the generative sequence with more nested hierarchies. As subsequences become more contextually dependent, the network responds by developing a greater number of distinct internal chunks. Through training, it gradually learns to reflect the underlying generative structure of the sequence, leading to a richer internal state space that disambiguates subtle contextual differences—representations that do not emerge without sufficient training.

---

> > ### Author Response · Authors · 2025-08-06
> >
> > Dear reviewer,
> >
> > Thank you for your constructive suggestions during the first review round of this paper, and we are looking forward to hearing your thoughts on our updated experiments and responses in the remaining two days in the author-reviewer discussion period. We are happy to answer additional concerns and questions promptly.
> >
> > We sincerely appreciate your time, consideration, and engagement throughout the review process.
> >
> > Best,
> > Authors of submission 6911

---

> > ### Comment · Reviewer_fcFE · 2025-08-08
> >
> > Thank you for the discussion about the new unsupervised algorithms for chunk discovery. I think the possibility of future insights from these unsupervised techniques is very interesting, especially as it offers additional perspectives from other popular unsupervised methods, e.g., SAEs.
> >
> > I am happy to see this paper appear at the conference, so I will raise my score.

---

### Official Review · Reviewer_Nmtb · 2025-06-30

**Clarity:** 2
**Significance:** 2
**Originality:** 2
**Rating:** 4
**Confidence:** 4

**Summary:**

This paper introduces a novel interpretability framework based on the “Reflection Hypothesis,” proposing that neural networks reflect structured regularities in their training data. The authors develop cognitively inspired chunking methods to extract recurring neural entities from RNNs and LLMs. Empirical results across architectures demonstrate that these extracted chunks correspond to meaningful concepts and can causally modulate model behavior.

**Questions:**

- What are the practical advantages of this chunking method over existing interpretability tools such as SAE?
For example, SAEs have been used for unlearning [4] and steering [5]. Can chunking be used similarly?
- Is the idea of chunking applicable beyond the language modality?


[4] Applying Sparse Autoencoders to Unlearn Knowledge in Language Models.
[5] A Survey on Sparse Autoencoders: Interpreting the Internal Mechanisms of Large Language Models.

**Ethical Concerns:**

["NO or VERY MINOR ethics concerns only"]

**Final Justification:**

My concerns have been mostly addressed, so I will raise my score to 4.

**Limitations:**

yes

**Quality:**

2

**Strengths And Weaknesses:**

### Strengths
The proposed Population Averaging (PA) method demonstrates superior performance in predicting concept occurrences compared to pretrained Sparse Autoencoders (SAEs), which is impressive. It is also a strong point that these results generalize across diverse model architectures, including Mamba and others.

### Weaknesses
- The core claim of the paper—the reflection hypothesis (i.e., that neural networks' internal representations reflect the structure of their training data)—does not appear particularly novel. Prior work such as [1] suggests that neural networks implicitly acquire syntactic structure; [2] shows that the geometric structure of SAE features reflects underlying data structure; and [3] demonstrates that cyclic structures in time-series data (e.g., weeks and months) are mirrored in internal representations. These studies collectively indicate that the reflection hypothesis is not new, and the contribution here may be more confirmatory than novel.

- The result in Figure 1(a), where replacing the hidden state of an RNN alters its output, seems rather expected—what is the new insight here?

- It is unclear why the authors use RNNs for toy tasks and Transformer-based LLMs for real-world scenarios. Why are there no toy experiments with Transformers to support consistency across architectures?

- In line 258, the paper states: “Figure 2a shows normalized activations of the first 50 neurons (unsorted) across all LLaMA3’s layers for the word ‘cheese’.” However, the result shown is from an RNN, not LLaMA3.

[1] Sudden Drops in the Loss: Syntax Acquisition, Phase Transitions, and Simplicity Bias in MLMs.
[2] The Geometry of Concepts: Sparse Autoencoder Feature Structure.
[3] Not All Language Model Features Are One-Dimensionally Linear.

---

> ### Author Rebuttal · Authors · 2025-07-30
>
> We are thankful for the reviewer’s suggestions. We are glad that the reviewer found our work “impressive”, raises a “strong point” and “generalizes across diverse model architectures”. We address the rest of the reviewer’s concerns, focusing on novelty, pragmaticity, and generalization to other modalities.
>
> > [W1] Prior work such as [1] … [2] … [3] …. indicate that the reflection hypothesis is not new, and the contribution here may be more confirmatory than novel.
>
> Thanks for pointing to the literature, we looked over these papers: [1] observed that structure onset corresponds to a sudden improvement of model linguistic ability. [2] analyze the low-D projected point cloud of SAE features and observe small-scale crystal-like, mid-scale modular, and large-scale power-law eigenvalue spectrum. [3] found multi-dimensional features in neural activities, and some of them correspond to circular structures in data, such as the day or the week.
>
> These are empirical findings and do not provide a theory on what causes their observations, in fact, the authors call for investigation on the origin to explain why neural data is structured in such a particular way.
>
> The reflection hypothesis postulates why we observe the phenomena in these papers, that is, a trained network may have its internal neural activities reflecting and mirroring the patterns in the data. This paper provides additional empirical support for the reflection hypothesis. We will integrate the suggested papers into the motivation for the hypothesis, and thank the reviewer for providing the suggestions.
>
> On a separate point, we would like to clarify that we do not view the reflection hypothesis as the primary novelty of this work.
>
> The main point we want to propose from this paper is a perception-grounded interpretability of artificial neural activities. Reviewer 86W2 noted it as “a fresh perspective beyond token-level or neuron-level probes”; fcFE “a novel framework for conducting future interpretability experiments”, ”grounded in cognitive science” which offers “ a human-in-the-loop approach”; 1jwn  “addresses a central challenge”, and “Introduces a compelling conceptual shift” and “aligns interpretability research more closely with cognitive neuroscience”.
>
> Specifically, instead of visualizing low-dimensional features from compute-heavy SAEs, which depend on the choice of the dimensionality reduction algorithms, we can leverage the chunking property of visual perception to directly discern entities and patterns in a subset of high-dimensional neural dynamics. We propose methods to find these reflections and show the role of these reflections on the network's computation. We tested the hypothesis and the chunk extraction methods from small-scale RNNs, to large-scale transformers, state-space models, and other large-scale networks with diverse architectures. Such general applicability of our method suggests that interpretability can be studied independently of training SAEs, freed from the variability induced by the choice of dimensionality reduction methods, and independently of the neural architectures under study.
>
> > [W2] The result in Figure 1(a), where replacing the hidden state of an RNN alters its output, seems rather expected.
>
> The hidden states are taken from the dictionary of state patterns extracted by the DSC method, while DSC extracts a dictionary of recurring population state patterns, we are not sure that this correlation contains a causation component and actually induces an alternation of network behavior in a predictable manner.  Therefore, we decided to graft the patterns to test whether it causally alters the network’s output in a predictable way.
>
> This experiment ensured us that the population activity patterns extracted in the dictionary actually causally influence the network’s behavior, in a predictable way, tightly coupled with the existing pattern in the training sequence. The causal component also implies that one can encourage the network to reuse by artificially altering the network’s neural state and prevent catastrophic forgetting during fine-tuning (Figure 1b, Supp. Figure 13). This experiment also precedes our motivation to perturb the network by grafting the extracted population activity states in the much more complex transformer-type scenarios.
>
> > [W3] It is unclear why the authors use RNNs for toy tasks and Transformer-based LLMs for real-world scenarios. Why are there no toy experiments with Transformers to support consistency across architectures?
>
> RNNs are a much simpler architecture than Transformers, allowing us to design a network with a small number of neurons without requiring hyperparameter choices of Transformers (e.g., positional embedding, number of heads, MLP hidden size) and with a simpler training recipe. We then use standardized large-scale architectures for a comparison between network types including RWKV (RNN), T5, and Llama-3 (Transformers). To ensure consistency across methods, we checked the effectiveness of PA and UCD in discovering the RNN chunks, and discussed them in Supp. section B.3, and Figure 8, 9. With this experimental setup, we are trying to ensure a line of consistency while demonstrating the generality of our claim across diverse architectures.
>
> > [Q1] What are the practical advantages of this chunking method over existing interpretability tools such as SAE? For example, SAEs have been used for unlearning [4] and steering [5]. Can chunking be used similarly?
>
> To clarify, we refer to unlearning as “freezing” and to steering as “grafting”, both of which we evaluate in our experiments. If the reviewer finds the terminology from the SAE literature more appropriate, we can update our paper accordingly.
>
> Our method for steering and unlearning is simpler and more direct than that of SAEs. In SAEs, steering is typically achieved by clamping a feature to a specific value—either through a constant multiplier or a scaled value relative to its maximum activation (Templeton et al., 2024), which probabilistically influences the network’s output during text generation.
>
> However, SAE features often do not correspond clearly to human-interpretable concepts. The mapping between features and words is indirect, requiring additional steps such as VocabProj or MaxAct to identify associated tokens [4]. This makes steering both complex and fragile. Interpreting SAE features remains an open challenge, and developing effective steering methods is still an active area of research [5].
>
> In contrast, our population-level chunking method offers more interpretable units by design. Chunks are derived from recurring patterns in the model’s activations, grounded in actual input semantics. This enables more transparent and effective interventions, such as concept grafting or unlearning, without the need for specialized heuristics. We also evaluated our steering efficiency on a bigger corpus across multiple contexts, and verified that the extracted chunks can be an efficient way of steering network behavior. More detailed results in a tabular form can be found in our discussion with reviewer 1jwn.
>
> The other is compute efficiency. The population averaging (PA) method is a post hoc, deterministic
> analysis that requires no optimization or gradient computation. Neural activations are precomputed, and the only cost lies in computing prototype vectors and evaluating distances across a small number of deviation thresholds—operations that scale linearly with the number of tokens and the embedding dimension and take on a scale of minutes without a GPU, even on production-level LLMs.
>
> In contrast, sparse autoencoders (SAEs) require multiple forward and backward passes through both encoder and decoder networks, typically over many training epochs. This results in orders-of-magnitude greater computational cost. PA, by contrast, offers a lightweight and accessible alternative for large-scale interpretability research.
>
> Our unsupervised chunk discovery method also remains computationally feasible. It runs efficiently on a single GPU. Given precomputed neural activations, it can be applied broadly to various architectures and datasets. A more detailed comparison of compute can be found in response to 86W2.
>
> > [Q2] Is the idea of chunking applicable beyond the language modality?
>
> Yes. We tested whether chunking could apply to vision by steering the vision component of a multimodal transformer (CLIP), which encodes both images and text into a shared latent space. To obtain the neural chunks for visual concepts, we used images with colored text (e.g., the word “RED” written in red or green). We extracted shape chunks from images containing consistent shapes of written words (e.g., “RED”, “BLUE”, 10 categories in total), and color chunks from applying PA to 10 images of uniform colors in each color category (10 categories). After extracting the color chunks and shape chunks, we applied chunk-based interventions to CLIP’s image embeddings. Specifically, given a set of congruent images (image of the shape RED written in red color), we tried grafting the chunk of a target color (e.g., red → blue) or target shape (e.g., “RED” → “BLUE”) to see if CLIP’s image embedding can be steered to align closer with the target concept than the source concept. These were tested on 90 curated examples (where font color matched the word). We measured effectiveness using cosine similarity shift: $\Delta = sim_{target, after} - sim_{source, after}$, where positive Δ indicates alignment with the target concept. We observed that grafting concept-encoding chunks in VLMs achieved 100% success for grafting shape chunks and 36.7% for color chunks. It suggests that neural subpopulations that encode visual concepts can be extracted and used to steer multi-modal models.

---

### Official Review · Reviewer_dfCF · 2025-07-02

**Clarity:** 3
**Significance:** 2
**Originality:** 3
**Rating:** 4
**Confidence:** 3

**Summary:**

This paper investigates the hypothesis, motivated by findings from human cognitive science, that artificial neural networks (ANNs) memorize recurring structural patterns in training data. The authors propose methods to identify such recurring patterns, or “chunks,” by clustering activation vectors. An activation vector $h \in \mathbb{R}^d$ is considered part of a chunk if its subvector $h_C$ lies within a threshold distance of a chunk centroid. For low-dimensional settings, chunk extraction is done by discretizing neuron activations into strings and forming a dictionary of recurring patterns. For large-scale networks, the authors use a supervised method, Neural Population Averaging (PA), to identify concept-specific chunks, and an unsupervised method that optimizes a dictionary to ensure each input is close to at least one entry. Through experiments on RNNs and transformer LLMs, they show that the extracted chunks correspond to interpretable and recurring input patterns, can be manipulated to affect model predictions, and capture hierarchical or contextual input structure.

**Questions:**

1. Please describe how the proposed chunking method compares to other concept discovery approaches (e.g., \[1–6]). What are the conceptual differences and potential advantages?
2. Provide a more formal and precise description of the Reflection Hypothesis.
3. Clearly define the terms *grafting* and *freezing* within the paper.
4. Explain the theoretical or empirical rationale behind the chunk clustering algorithm. Which theory, if any, guided the algorithm design choices?

**Ethical Concerns:**

["NO or VERY MINOR ethics concerns only"]

**Final Justification:**

I believe this paper makes a valuable contribution to the interpretability field by introducing a novel concept discovery method grounded in cognitive science. While I initially had concerns about the lack of discussion of closely related work, missing key definitions, and insufficiency of systematic analysis, I appreciate that the authors have addressed these points during the rebuttal. I am hopeful that these improvements will be reflected in the final version of the manuscript.

**Limitations:**

The authors adequately addressed the limitations.

**Paper Formatting Concerns:**

* No line numbers and seemingly incorrect line spacing in paragraphs between lines 130-131 and 156-157.
* Some equations, e.g., between lines 156-157, 158-159, are not numbered.

**Quality:**

3

**Strengths And Weaknesses:**

**Strengths**

* The paper is generally well-written and clearly structured.
* It presents a novel perspective on interpretability, drawing inspiration from human cognitive science, which adds an interesting conceptual framing to the work.
* The proposed method has demonstrated utility in downstream applications, by enabling causal interventions on model outputs through chunk manipulation.


**Weaknesses**

* While the work aligns with the broader literature on concept-based interpretability and mechanistic interpretability, the related work section lacks coverage of key concept discovery methods beyond sparse autoencoders (SAE). Additionally, these competing methods are not included in the quantitative evaluation \[1–6].
* Most quantitative results represent an exploratory analyses rather than a rigorous, systematic evaluation.
* There are several clarity issues throughout the paper:

  1. The Reflection Hypothesis is introduced in a vague and imprecise manner.
  2. It is not stated whether a single input can be associated with multiple chunks.
  3. Key terms such as *grafting* and *freezing* are not properly introduced or formally defined.
* The decision process behind the chunking strategies in Section 3.1 is not thoroughly explained. The overall methodology appears somewhat ad hoc and lacks strong theoretical grounding.

**Minor Comments:**

* Missing space after “(SAEs)” (line 69), “at the time of the concept-specific index” (lines 156–157), and “Figure9” (line 282), double period typo in “neural activities..” (line 234).
* Figure 2a (line 258) reference does not point to the correct figure.
* Figure 3 (left side): unclear how different tokenizations are represented.
* Some variables are not defined or are ambiguous:

  * $\overline{h_C}$ (line 128)
  * $h_1, \ldots, h_d$ (lines 130–131)


References:

[1] Ghorbani, A., Wexler, J., Zou, J.Y. and Kim, B., 2019. Towards automatic concept-based explanations. Advances in neural information processing systems, 32.

[2] Vielhaben, J., Bluecher, S. and Strodthoff, N., Multi-dimensional concept discovery (MCD): A unifying framework with completeness guarantees. Transactions on Machine Learning Research.

[3] Fel, T., Boutin, V., Béthune, L., Cadène, R., Moayeri, M., Andéol, L., Chalvidal, M. and Serre, T., 2023. A holistic approach to unifying automatic concept extraction and concept importance estimation. Advances in Neural Information Processing Systems, 36, pp.54805-54818.

[4] Black, S., Sharkey, L., Grinsztajn, L., Winsor, E., Braun, D., Merizian, J., Parker, K., Guevara, C.R., Millidge, B., Alfour, G. and Leahy, C., 2022. Interpreting neural networks through the polytope lens. arXiv preprint arXiv:2211.12312.

[5] Tamkin, A., Taufeeque, M. and Goodman, N., 2024, July. Codebook Features: Sparse and Discrete Interpretability for Neural Networks. In International Conference on Machine Learning (pp. 47535-47563). PMLR.

[6] Kim, B., Wattenberg, M., Gilmer, J., Cai, C., Wexler, J. and Viegas, F., 2018, July. Interpretability beyond feature attribution: Quantitative testing with concept activation vectors (tcav). In International conference on machine learning (pp. 2668-2677). PMLR.

---

> ### Author Rebuttal · Authors · 2025-07-30
>
> We thank the reviewer for the constructive suggestions. Below we address the reviewer’s comments, focusing on clarifying the literature, quantitative results, and supplementary definitions.
>
> > [W1, Q1] While the work aligns with the broader literature on concept-based interpretability and mechanistic interpretability, the related work section lacks coverage of key concept discovery methods beyond sparse autoencoders (SAE). Additionally, these competing methods are not included in the quantitative evaluation [1–6].
> ... Please describe how the proposed chunking method compares to other concept discovery approaches (e.g., [1–6]). What are the conceptual differences and potential advantages?
>
> We thank the reviewer for pointing to related concept-based literature, for which we will add a discussion in the related works section. However, most of the mentioned papers study concept-based interpretability methods for image classifiers, sometimes assuming accessibility to a "class" label. In Ghorbani et al. [1], Fel et al. [3] and Kim et al.[6], concepts correspond to visual features in the image dataset (e.g. pixels of a wheel), and they study what part of the image influences the network's classification decision. Black et al. [4] define polytopes as linear regions in input space and study the geometry of the input space. Vielhaben et al. [4] define concepts as a linear subspace in the feature space, spanned by a number of basis vectors that is unbounded. Finally, while Tamkin et al. [5] learns a codebook of features, their method proposes training a model with codebook bottlenecks requiring significantly more compute than our methods. Moreover, we intend to interpret the raw activity of large models regardless of their training procedure or architectures instead of optimizing a new model.
>
> In contrast, our work primarily studies network interpretability processing sequences of tokens. Our work examines how the internal structure of the network exhibits recurring patterns of entities. We reframe interpretability by connecting neural population trajectories to human chunking.This approach to interpretability was noted by reviewer 86W2 as “a fresh perspective beyond token-level or neuron-level probes”; fcFE “a novel framework for conducting future interpretability experiments”, ”grounded in cognitive science” which offers “ a human-in-the-loop approach”; 1jwn “addresses a central challenge”, and “Introduces a compelling conceptual shift” (1jwn).
>
> While we agree on the importance of these comparisons, the aforementioned differences make a quantitative comparison a non-trivial problem and out of scope for the rebuttal.
>
> > [W2] Most quantitative results represent an exploratory analysis rather than a rigorous, systematic evaluation.
>
> We want to make a point about there existing recurring neural population activity patterns in sequence processing models, in response to concrete and abstract patterns in the input sequences, and these neural patterns influence the network behaviors. Due to the nature of this argument, it is not a message that can be easily conveyed by quantitative measures and demands ample examples to support. But we tried to strike a balance between exploratory and quantitative results by testing our claim across architectures (Mamba, RNNs, state space models, T5 in Figure 5, Supp. Figure 24-26); inventories (beyond simple examples to top 100 english words in Figure 5, Supp. Figure 21-24 and abstract POS tags in Figure 31), and corpus (from topic related sentences to sentences with an underlying abstract structure, to literary corpus, in Figure 5, 6).
>
> Apart from that, we supplemented the evaluation of grafting effectiveness by quantitative evaluation to the ROCStories benchmark, sampling 2,000 sentences to extract neural population chunks corresponding to the top 20 frequent concept words (filtered to avoid short or overlapping terms) in addition to the existing concepts collected. To assess context-dependent effects of chunk grafting, we used prompts from the TREC Question Classification dataset, which includes six coarse-grained context categories. For each category, we sampled 50 prompts and evaluated grafting success by measuring the difference in concept occurrence probability between grafted and ungrafted (control) LLaMA-3 generations. For example, a +41% increase indicates that the target concept is 41% more likely to appear in the generating output with grafting compared to generating output without grafting.
>  To reduce computational cost, we categorized grafting into early (layers 1–9), middle (10–19), and late (20–29) interventions. We found that grafting consistently increased concept occurrence across contexts, with early-layer grafting being the most effective. More details can be found in our discussion with reviewer 1jwn.
>
> | TREC Category | Early Δ   | Middle Δ  | Late Δ   |
> |---------------|-----------|-----------|----------|
> | ABBR          | +41.0%    | +15.9%    | +3.1%    |
> | DESC          | +33.4%    | +12.5%    | +5.1%    |
> | ENTY          | +35.5%    | +9.9%     | +4.3%    |
> | HUM           | +34.8%    | +9.6%     | +3.3%    |
> | LOC           | +36.8%    | +9.8%     | +3.7%    |
> | NUM           | +33.8%    | +10.3%    | +4.5%    |
>
> > [W3] The Reflection Hypothesis is introduced in a vague and imprecise manner.… Provide a more formal and precise description of the Reflection Hypothesis.
>
> Let the sequence be denoted as $\( S = (s^1, s^2, \dots, s^n) \)$, indexed by $\( I = \{1, 2, \dots, n\} \)$. A pattern in the input is defined as a recurring subsequence carrying the same concept
> $$P = (s^{i}, s^{i+1}, \dots, s^{i+k-1}) \subseteq S $$
> of fixed or variable length $k$, where the same sequence of tokens appears many times in \( S \).
>
> The _Reflection Hypothesis_ posits that such recurring input patterns $ P $ are mirrored in the internal neural dynamics of the model, such that their corresponding neural activations
> $(\mathbf{h}^{i}, \mathbf{h}^{i+1}, \dots, \mathbf{h}^{i+k-1})$
> lie within a population-level chunk defined as a ball $\overline{B}(\overline{\mathbf{h}}_C, \Delta) \subset \mathbb{R}^d$
> centered at a prototypical activation $\overline{\mathbf{h}}_C$ with radius $ \Delta $.
>
> Regularities in the input sequence are reflected as localized activity patterns in the neural activities.
>
> > [ W4] It is not stated whether a single input can be associated with multiple chunks.
>
> A single token can be associated with multiple chunks, if this input has a different meaning as in different contexts. For example, the ‘ese’ token may elicit a different chunk in the network when it is used to parse ‘Che’+’ese’ from when it is used to parse ‘Ge’ +’ese’.
>
> > [ W5] Key terms such as grafting and freezing are not properly introduced or formally defined.
>
> __Grafting__ refers to the targeted insertion of previously learned chunks into a neural network’s internal computation. We modify its internal activations $ \mathbf{h} $ at a specified layer $ l $ and neural subset $ C $ by grafting the chunk $\mathbf{h}_C \leftarrow \overline{\mathbf{h}}_C$ to replace the original neural subpopulation activities.
>
> __Freezing__ refers to the operation of zeroing out the neural activations $ C $ that support the chunk. This is implemented by replacing the network activation with stored chunk representations at the corresponding neurons. At a given layer $ l $, we enforce: $\mathbf{h}_{C,l} = 0.$
>
> > [W6] Explain the theoretical or empirical rationale behind the chunk clustering algorithm. Which theory, if any, guided the algorithm design choices?
>
>
> We hypothesize that there exist recurring concepts as patterns in the input $P = (s^{i}, s^{i+1}, \dots, s^{i+k-1}) \subseteq S $, $\( S = (s^1, s^2, \dots, s^n) \)$. These concepts contain tokens that elicit similar context and distributions (the definition of a word in the dictionary of a language). This follows similar logic to distributional semantics (e.g., word2vec, GloVe), where the mean of many contextual embeddings approximates a central semantic prototype.
>
> The _Reflection Hypothesis_ posits that such recurring input patterns $ P $ are mirrored in the internal neural dynamics of the model, such that their corresponding neural activations
> $(\mathbf{h}^{i}, \mathbf{h}^{i+1}, \dots, \mathbf{h}^{i+k-1})$ lie within a population-level chunk defined as a ball $\overline{B}(\overline{\mathbf{h}}_C, \Delta) \subset \mathbb{R}^d$ centered at a prototypical activation $\overline{\mathbf{h}}_C$ with radius $ \Delta $.
> When two sequences carry the same concept, we expect them to elicit similar activities for a subpopulation of neurons within the network.
>
> In our larger-scale experiments, we observe that PA and UCD found neural population clusters and correlate well with concrete and abstract concepts in data from recurring words, to sentence structure, to POS tags, suggesting that for a substantial number of concepts, a stable signal emerges. We will supplement more theoretical grounding in the paper.

---

> > ### Comment · Reviewer_dfCF · 2025-08-03
> >
> > I thank the authors for their thorough and constructive rebuttal. Most of my concerns have been addressed. With the planned additions to the revised version, including the extended related work section, the clarified definition and theoretical grounding, and the new experimental results, I am increasing my score to borderline accept.

---

> > > ### Author Response · Authors · 2025-08-06
> > >
> > > Thank you so much for your thoughtful follow-up and for helping us improve this work. We're really glad to hear that our clarifications and updates addressed your concerns. We truly appreciate your reconsideration and will make sure to incorporate the planned revisions into the final version if the paper is accepted.

---

### Official Review · Reviewer_1jwn · 2025-07-03

**Clarity:** 3
**Significance:** 2
**Originality:** 3
**Rating:** 5
**Confidence:** 3

**Summary:**

The authors propose the Reflection Hypothesis - the idea that the raw neural population activity of neural networks exhibits patterns that reflect the statistical regularities of the training data. To interpret these dynamics, the authors propose several cognitively-inspired 'chunking' methods to segment high-dimensional neural trajectories into recurring, interpretable units. These include Discrete Sequence Chunking (DSC) when $d$ is small (low-dimensional RNNs), Neural Population Averaging (PA) when $d$ is very large (transformers), and Unsupervised Chunk Discovery (UCD) when labels are unavailable. Interestingly, across architectures (RNNs and LLaMA-3), the authors demonstrate that these methods recover recurring internal states aligned with concrete and abstract input concepts. Furthermore, intervention-style experiments (e.g., grafting neurons to discovered chunks) demonstrate that some discovered chunks have a systematic causal effect on model behaviour, providing evidence that chunks are not just spurious features, but correspond to *functionally* meaningful internal representations.

**Questions:**

Clarification on chunk minimality: I'm curious if the discovered chunks are minimal, in the sense of being irreducible functional units. E.g., could the chunk for 'cheesecake' be decomposable into smaller sub-chunks ('cheese' and 'cake') with independent causal effects?

Justification for PA methodology: the PA method assumes averaging hidden states across multiple contexts yields a meaningful, general 'chunk' for the concept. Could the authors maybe provide some evidence or justification for this assumption, especially given the context-sensitivity of internal representations in transformers?

Quantitative evaluations: the paper presents compelling qualitative demonstrations to implicate chunks in the functional behaviour of larger models, but offers limited large-scale quantitative evaluation of chunk causal efficacy. Have the authors considered more systematic evaluation strategies - e.g., measuring the consistency of chunk-induced output changes across a large, diverse dataset?

Unsupervised setting and chunk evaluation: in this setting, how do the authors determine whether learned chunks are semantically distinct or functionally redundant? Related to this, have the authors thought about any systematic way to assess the 'quality' of a chunk? Or what determines a 'good' chunk, e.g., semantically distinctness, functionally consistency, minimality?

**Ethical Concerns:**

["NO or VERY MINOR ethics concerns only"]

**Final Justification:**

The authors provided a detailed and thoughtful rebuttal that addressed many of my initial concerns. The new results on temporal dynamics, context sensitivity, and chunk evaluation significantly strengthen the empirical contribution. The additional grafting experiments and large-scale evaluations also lend support to the claim that discovered chunks are functionally meaningful.

That said, some limitations remain, particularly regarding the reliance on early layers, a focus on concrete concepts, and open questions about the robustness of chunk functionality across architectures and linguistic categories. These do not undermine the core contributions, but they do suggest the need for further work to fully establish generality.

Overall, the paper introduces a conceptually interesting and potentially impactful framework, and the authors have made a compelling case for its validity. I maintain my positive recommendation.

**Limitations:**

I would like to see potentially some discussion on the context-agnostic assumptions underlying the PA and UCD approaches, or a broader discussion about this in the supplementary, perhaps.

**Paper Formatting Concerns:**

No concerns.

**Quality:**

2

**Strengths And Weaknesses:**

**Strengths**:
- The paper addresses a central challenge in modern deep learning - how to interpret the internal structure and reuse of representations in large models.
- Introduces a compelling conceptual shift in how we approach neural interpretability. Rather than focusing on individual neurons or attention heads, the authors argue that the dynamics of the neural population should be the primary unit of analysis. Crucially, the authors also challenge the dominant focus on linear decodability (e.g., probing), and instead emphasises functional and causal efficacy of the units of analysis. This reframing aligns interpretability research more closely with cognitive neuroscience, where population dynamics are a central unit of analysis.
- Methodological breadth: the paper introduces three chunking methods for use across different settings (supervised vs unsupervised, low/high-dimensional settings), making the framework quite applicable to many scenarios.
- The use of causal interventions (e.g., chunk grafting) demonstrate the chunks are not just correlated with behaviour, but causally implicated.  This kind of interventional evaluation strengthens the paper's empirical claims.
- Architecture agnostic evaluation: chunking framework validated across diverse architectures, supporting the claim that chunk-like internal structure is a general emergent property arising from statistical regularities of the training data, rather than any specific model architecture.

**Weaknesses**
- Chunking methods are insensitive to temporal dynamics: only one of the three proposed methods (DSC) explicitly incorporates temporal information by merging adjacent symbols in the sequence. The other two methods are snapshot based, which effectively ignores the trajectory-level structure that is essential to sequential computation (especially in models e.g., transformers processing long sequences).
- Context invariance assumption for PA: assuming that a concept can be represented by a single population vector obtained by averaging across occurrences may be invalid in context-sensitive models (e.g., a concept like 'cat' could produce different neural activations depending on its context, and averaging across all instances could obscure the underlying structure). This compounds the issue of using static snapshots, since the temporal and contextual dependencies are not preserved in the averaged chunk.
- Evaluation: for the LLM case, there are only qualitative examples (e.g., for grafting and freezing) instead of quantitative metrics in the interventional test to causally implicate chunk identity in the model's functional behaviour. This makes difficult to assess whether the claim that chunks are functionally useful actually generalises to these larger models. Similarly, in the LLM case, for quantitative metrics (e.g., decoding in Fig 3), the evaluation setting is quite narrow (3 words), again making it difficult to assess whether the chunk behaviours are exceptions or consistent patterns that hold for larger models.
- Lack of generalisation analysis for chunks: Relating to point 2, while chunks are learned in a context-agnostic way, it is unclear that the chunks can be consistently reused across different contexts. I would be curious to see some evaluation of a learned chunk's (e.g., 'cheese') consistency across different contexts, perhaps by e.g., grafting the chunk across multiple contexts and quantitatively measuring that the impact across all contexts is consistent.

---

> ### Author Rebuttal · Authors · 2025-07-30
>
> We thank the reviewer for the very thoughtful and constructive feedback. We are encouraged by the recognition that the work “addresses a central challenge,” “introduces a compelling conceptual shift,” and demonstrates “methodological depth.” Please see our responses to the comments and questions below, focusing on clarification on the temporal dynamics, additional quantitative measures, and contextual sensitivity.
>
> > [W1] Chunking methods are insensitive to temporal dynamics ...
>
> We would like to clarify that there is indeed a temporal dynamics extracted by both PA and UCD, but the paper only illustrates snapshots to show the smallest chunk granularity. PA is not limited by snapshots, Figures 18 and 19 in Supp. shows the temporal dynamics of PA before and after the temporal snapshot for the illustrated examples. Figure 20 in Supp. illustrates the temporal dynamics preceding and proceeding concepts for other examples, Figures 21 and 22, from pages 23 - 25, evaluate the extracted PA chunks with a temporal component.
>
> For UCD, condensing neural activity into discrete chunks enables the DSC algorithm to identify underlying temporal chunk patterns. There are rich temporal patterns of chunk interactions—for instance, in the Emma corpus at layer 10, the chunk sequence 1145-136-1300-367 occurs 80 times, and the related pattern 1145-1300-367-367-367 appears 115 times—indicating that neural computations often span multiple tokens.
> We quantify this by calculating the compression ratio, the number of tokens in the corpus divided by the number of extracted temporal chunks that span the same corpus. This suggests that beyond the level of tokens, there are also neural computation dynamics described by the chunks extracted.
>
> | Layer             | 5    | 10   | 15   | 20   | 25   | 30   |
> |-------------------|------|------|------|------|------|------|
> | Compression Ratio | 0.39 | 0.45 | 0.62 | 0.55 | 0.43 | 0.63 |
>
>
> > [W2] For grafting and freezing, quantification of whether the functionality of chunks generalizes to larger models?
>
> We measured causal effectiveness by comparing how often a concept appears in LLM-generated text with and without intervention (grafting or freezing).
>
> ### Grafting
> **Prompt:** “Hi, how are you doing?”
> **N = 100**
>
> | Target Concept | Without Grafting | With Grafting |
> |----------------|------------------|----------------|
> | cake           | 1%               | 83%            |
> | cheese         | 0%               | 90%            |
> | cheesecake     | 0%               | 16%            |
>
>
> ---
>
> ### Freezing
>
> **Prompt:** “What is the name of a rich, savory food made from curdled milk, often aged to enhance its flavors?”
> **Target concept to freeze:** cheese
> **N = 100**
>
> | Condition    | Cheese Occurrence |
> |--------------|-------------------|
> | No Freezing  | 67%               |
> | Freezing     | 33%               |
>
> **Prompt:** “Name three most common desserts in Europe.”
> **Target Concept to freeze:** cake
> **N = 100**
>
> | Condition    | Cake Occurrence |
> |--------------|-----------------|
> | No Freezing  | 36%             |
> | Freezing     | 9%              |
>
> Overall, grafting was more effective than freezing, though both had an effect.
>
> > [W3] Similarly, in the LLM case, for quantitative metrics (decoding), the evaluation setting is quite narrow (3 words), again making it difficult to assess whether the chunk behaviours are exceptions or consistent patterns that hold for larger models.
>
> Beyond the three illustrated word examples, we extended our evaluation to the top 100 most frequent English words in a larger corpus (Figure 5) and applied the same quantitative metrics across four large models with distinct architectures. In addition to LLaMA-3, we tested Mamba (a state space model), T5 (an encoder-decoder model), and RWKV (a large-scale RNN). We also assessed the model’s ability to extract chunks corresponding to abstract structural schemas (Figure 4). Across these models, we found that the discovered neural chunks consistently and deterministically predicted concept occurrence in held-out test corpora, often achieving 100% decoding accuracy across a number of early layers ( see Supp. Figures 24, 25, and 26) .
>
> > [W4] Lack of generalisation analysis for chunks: ... it is unclear that the chunks can be consistently reused across different contexts, … e.g., grafting the chunk across multiple contexts and quantitatively measuring that the impact across all contexts.
>
> We extended our quantitative evaluation to the ROCStories benchmark (Mostafazadeh et al., 2016), a narrative corpus of coherent story sentences, and assessed grafting consistency across context categories using the TREC Question Classification dataset (Li and Roth, 2002). From ROCStories, we sampled 2,000 sentences and expanded the existing PA extracted chunks by adding the 20 frequent words (≥5 characters) in the corpus.
>
> To evaluate context-dependency, we sampled 50 prompts from each TREC category—ABBR, DESC, ENTY, HUM, LOC, and NUM—and measured the percentage success of grafting target concepts in each. To reduce computational load, we grouped grafting interventions into early (layers 1–9), middle (10–19), and late (20–29) layers, and compared concept occurrence probabilities between grafted and control generations.
>
> | TREC Category | Control        | Early Graf.     | Mid Graf.       | Late Graf.      |
> |---------------|----------------|------------------|------------------|------------------|
> | ABBR          | 14.9% ± 24.6%  | 55.9% ± 32.4%   | 30.8% ± 26.9%   | 18.0% ± 25.8%   |
> | DESC          | 15.6% ± 21.5%  | 49.0% ± 26.8%   | 28.1% ± 22.4%   | 20.7% ± 23.0%   |
> | ENTY          | 12.6% ± 21.8%  | 48.1% ± 28.6%   | 22.5% ± 24.1%   | 16.9% ± 22.9%   |
> | HUM           | 11.9% ± 20.0%  | 46.7% ± 28.6%   | 21.5% ± 23.8%   | 15.2% ± 20.7%   |
> | LOC           | 10.7% ± 19.4%  | 47.5% ± 28.6%   | 20.5% ± 22.4%   | 14.4% ± 21.2%   |
> | NUM           | 11.5% ± 18.4%  | 45.3% ± 27.0%   | 21.8% ± 21.4%   | 16.0% ± 20.9%   |
>
> Grafting concept-related chunks increases the likelihood of LLaMA-3 generating the target concept, with little variation across contexts. Grafting early-layer is more effective than in later layers.
>
> > [Q1] Any systematic way to assess the 'quality' of a chunk? Or what determines a 'good' chunk, e.g., semantically distinctness, functional consistency, minimality?
>
> Evaluation in the unsupervised setting was a concern for us, and we explored several measures to assess whether the algorithm achieved its intended outcomes.
>
> A good chunk should closely approximate the embedding. We compute the normalized cosine similarity between each embedding and its best-matching chunk in the dictionary, and ensure the similarity scores follow a reasonable distribution.
>
> We also check semantic distinctness, i.e., each embedding should be similar to only a few chunks while remaining dissimilar to most others in the dictionary. Figure 27b shows the similarity between a single embedding and the set of learned chunks in layer 10 of LLaMA.
>
> Additionally, we verify that chunk usage is reasonably distributed—that is, no small subset of chunks should dominate as the most similar matches across embeddings. This diversity in chunk identification is illustrated in Figure 27c. W
>
> e also visually inspect samples of extracted chunks alongside two matching embeddings and a non-matching embedding as a control. Example sets are shown in Figure 28. Further discussion on chunk evaluation is provided in Section E of the Supp.
>
> > [W2, Q2, L] PA assumes that averaging hidden states across multiple contexts yields a meaningful, general 'chunk' for the concept... Could the authors provide some justification for this assumption, given the context-sensitivity of internal representations in transformers?
>
> We appreciate the reviewer’s insightful comment regarding the context invariance assumption underlying PA and UCD. Indeed, modern language models are known to exhibit context-sensitive internal representations, and a given word (e.g., "bank") may evoke distinct activations depending on its context (e.g., a river bank versus a financial bank).
>
> However, our goal is not to model context-specific usage, but rather to extract the core commonalities a concept elicits in neural computation activities across contextual variations —analogous to semantic prototypes in word2vec and GloVe.
>
> This assumption on the algorithm then finds its empirical validation. We observe that PA and UCD-derived chunks correlate with many concepts. This suggests that for at least a subset of words, there exists a stable commonality despite contextual variations. And for other words, such as prepositions, which are contextually more nuanced, we would expect this method to be less effective.
>
> Indeed, we observed that the grafting success of PA-derived chunks depends on the type of word. Grafting proved especially effective for concrete concepts. For example, inserting the chunk for "school" increased its occurrence from 0.67% in control generations to 78.7% with grafting; "Christmas" rose from 0.33% to 74.3%, and "family" from 1.7% to 71.3%. For non-specific words such as prepositions and conjunctions, grafting PA chunks tends to be less effective: inserting the chunk for “if” only increased its occurrence from 6.7% to 7.3%, “at” from 7.7% to 11.7%, and “on” from 23.0% to 28.7%. This limited impact is often correlated with lower PA chunk quality, which is also reflected in the decoding evaluation metrics for these function words.
>
> > [Q4] Could the chunk for 'cheesecake' be decomposable into smaller sub-chunks ('cheese' and 'cake')... ?
>
> These representations do not reduce to distinct functional subgroups. In the examples shown, over 99% of the neurons encoding cheese, cake, and cheesecake overlap, with only 11, 2, and 34 neurons uniquely encoding each, respectively. This suggests that the same neural population supports multiple meanings.

---

> > ### Author Response · Authors · 2025-08-06
> >
> > Dear reviewer,
> >
> > Thank you for your constructive suggestions during the first review round of this paper, and we are looking forward to hearing your thoughts on our updated experiments and responses in the remaining two days in the author-reviewer discussion period. We are happy to answer additional concerns and questions promptly.
> >
> > We sincerely appreciate your time, consideration, and engagement throughout the review process.
> >
> > Best,
> > Authors of submission 6911

---

> ### Comment · Reviewer_1jwn · 2025-08-07
>
> I thank the authors for their thorough and thoughtful rebuttal. They have clearly made a strong effort to address the concerns raised, and the additional experiments (e.g. on grafting consistency, decoding generalisation, chunk evaluation) substantially strengthen the empirical side of the paper. I especially appreciated the more detailed justification of the PA method and the exploration of chunk quality in the unsupervised setting.
>
> That said, a few concerns remain. While the rebuttal demonstrates that discovered chunks often correlate with concept representations and can be functionally manipulated via grafting, the assumption that context-invariant neural prototypes are sufficient for interpretability remains somewhat under-examined. In particular, PA and UCD still collapse across contexts in a way that may obscure fine-grained structure - especially for abstract or polysemous concepts. The analogy to word2vec-style prototypes is helpful, but does not fully address the risks this averaging poses for downstream interpretation or causal inference.
>
> Additionally, the new results on chunk generalisation and context robustness are encouraging, and suggest that functional chunks may transfer across diverse contexts and architectures. That said, these effects appear strongest in early layers and for certain word types (e.g., concrete nouns), so questions remain about how reliably chunk functionality generalises across layers, linguistic categories, and broader settings.
>
> Overall, I continue to view this paper as a compelling and original contribution, with meaningful conceptual and methodological innovations. The rebuttal successfully addressed many concerns, though some questions around context sensitivity and the generality of causal effects remain partially open. My evaluation remains positive, I will raise my score by 1, and I maintain my recommendation for acceptance.

---

> > ### Author Response · Authors · 2025-08-08
> >
> > We are deeply grateful to the reviewer for the recommendation and for recognizing the significance and innovation of this work. We believe that the method introduced here offers a promising step toward incorporating principles of cognition with interpreting neural population dynamics. And we are encouraged by its effectiveness.
> >
> > Building on the foundation set by this work, we plan to explore further chunk sensitivity, inter-layer processing mechanisms, and their dependency with abstract/concrete concepts in our upcoming investigations, and we look forward to sharing them with the community.

---

### Official Review · Reviewer_86W2 · 2025-07-03

**Clarity:** 3
**Significance:** 3
**Originality:** 3
**Rating:** 5
**Confidence:** 3

**Summary:**

This paper introduces a novel framework for interpreting neural network computations through the cognitive lens of chunking, proposing the Reflection Hypothesis: well-trained neural networks exhibit internal neural population patterns that mirror the regularities in their training data. The authors develop three complementary methods—Discrete Sequence Chunking (DSC), Neural Population Averaging (PA), and Unsupervised Chunk Discovery (UCD)—to extract recurring, interpretable neural activity patterns (chunks) from both small RNNs and large language models (LLMs). These chunks capture concrete or abstract concepts present in input sequences. The authors empirically validate the hypothesis through comprehensive experiments spanning synthetic RNN tasks and modern LLMs like LLaMA-3, showing that chunk activations robustly align with recurring words and abstract narrative schemas. Grafting and freezing these chunks causally affect the models’ predictions, offering mechanistic interpretability grounded in cognitively inspired principles.

**Questions:**

1. How do the authors envision chunk discovery scaling to production-sized LLMs (e.g., 70B+) given the need for large embedding datasets and iterative dictionary learning?

2. Could chunk-based interpretability complement or enhance existing sparse autoencoder approaches or linear probe techniques? Any preliminary experiments combining them?

3. Have the authors explored whether the extracted chunks remain stable or shift when the same base model is finetuned on different tasks? This could reveal insights into concept drift during finetuning.

4. Beyond the included narrative schema experiments, could future work assess whether chunk activations correlate with human attention or comprehension on tasks like QA or summarization?

**Ethical Concerns:**

["NO or VERY MINOR ethics concerns only"]

**Limitations:**

Yes. The authors clearly acknowledge the potential dependency on training data structures, the challenges in chunking highly dynamic contexts, and limitations of relying solely on chunk-based explanations for interpreting complex behaviors. They also highlight promising directions for future research to formalize theoretical underpinnings, which aligns with best practices in interpretability research.

**Paper Formatting Concerns:**

None.

**Quality:**

3

**Strengths And Weaknesses:**

Strengths:

- The Reflection Hypothesis reframes interpretability by connecting neural population trajectories to human-like chunking, a fresh perspective beyond token-level or neuron-level probes.

- The three proposed chunk extraction techniques effectively handle a wide range of settings, from low-dimensional RNNs to high-dimensional LLMs.

- The paper presents extensive experiments demonstrating chunk consistency across architectures, scalability to LLMs, and the causal influence of extracted chunks on model behavior.

- The ability to graft or freeze identified chunks and observe systematic changes in model output provides compelling evidence of interpretability.

- Validation on diverse architectures (RNNs, transformers, state-space models) supports the robustness and applicability of the proposed methods.

Weaknesses:

- The paper could better contextualize how its framework integrates with or contrasts against recent neuron-centric interpretability methods like SAE circuits or representation engineering.

- Some details on the computational cost of applying chunk extraction at scale in large models are missing.

- While synthetic and narrative-driven datasets are useful, real-world downstream tasks (e.g., factuality or bias assessment) would further strengthen practical significance.

---

> ### Author Rebuttal · Authors · 2025-07-30
>
> We thank the reviewer for your insightful suggestions. We clarify and contrast our method with alternative interpretability methods a bit more, especially on computation. We also ran more evaluations on the grafting experiment and examined the grafting effectiveness in multi-modal models.
>
> > Some details on the computational cost of applying chunk extraction at scale in large models are missing.
>
> All interpretability methods that analyze neural population activities need access to LLM neural activities, which can be acquired in a single forward pass on the data. This is required for SAE, representation engineering, and chunking methods alike.
>
> For PA that extracts chunks from shorter input, taking a 1,000-token sequence for example, our method is lightweight. The cosine-similarity-based matching and tolerance sweep are applied to hidden state arrays of ~16MB per layer, resulting in total memory usage under 200MB. Even across all 32 layers, the complete decoding procedure finishes in under 1 minute on a standard CPU. This makes our analysis tractable even in constrained environments.
>
> The unsupervised chunk learning model consists of ~8.2 million parameters (learning a matrix of shape [2000, 4096]) and operates primarily through matrix multiplication for cosine similarity computation. With input sequences comprising ~200,000 tokens (e.g., Austen–Emma), the batch-based training regime (batch size 32) requires under 100MB of GPU memory per step. On a single NVIDIA RTX 3090, training for 1 epoch takes 30-60 seconds, and 100 epochs takes about 1-2 hours. All layers will take 32-64 hours if done sequentially. In comparison to the SAE variate, where the latent layer has 131,072 neurons (i.e., 32× the input dimension of 4096). This increases the parameter count to ~1.07 billion. The same setting completes 100 epochs in approximately 6–10 hours per layer. All layers will take 192-320 GPU-hours.
>
> We will add a discussion of these computational costs, along with concrete runtime estimates and scaling behavior in the supplementary material to clarify the tractability of our approach at scale.
>
> > How do the authors envision chunk discovery scaling to production-sized LLMs (e.g., 70B+) given the need for large embedding datasets and iterative dictionary learning?
>
> When applied to a production-scale model like LLaMA-70B, our methodology remains tractable. For PA over sequences such as one containing 1000 tokens, each hidden state array occupies around 32MB, and the total memory usage across all 80 layers is approximately 2.5GB if done simultaneously. The total runtime will be a few minutes on CPU per concept across all layers. The unsupervised chunk learning model, which learns a 2000 × 8192 matrix (around 16.4 million parameters), will require under 1GB of GPU memory per step and trains in approximately 2–3 minutes per epoch, totaling 3–5 hours for 100 epochs on a single NVIDIA RTX 3090, with batch size 32 and 200,000 input tokens. All 80 layers shall take 240 - 400 GPU-hours.  In contrast, the sparse autoencoder variant with a 262,144-dimensional latent layer (32× overcomplete) increases the parameter count to around 4.1 billion, requiring around 50GB of memory and 12–20 hours to train over 100 epochs per layer. All 80 layers will take 860 - 1600 GPU-hours.
>
>
> > While synthetic and narrative-driven datasets are useful, real-world downstream tasks (e.g., factuality or bias assessment) would further strengthen practical significance.
>
> We appreciate the reviewer’s suggestion and agree that evaluating in real-world contexts is important. To this end, we extended our analysis to the ROCStories benchmark—a naturalistic narrative corpus—and evaluated grafting effectiveness using the TREC Question Classification dataset, which spans six real-world categories. For each category, we sampled 50 prompts and compared generation outcomes with and without grafting. We noticed that averaging all collected words, grafting consistently improved word occurrence probability across all context categories: +41% (ABBR), +33% (DESC), +36% (ENTY), +35% (HUM), +37% (LOC), and +34% (NUM). These results highlight the method’s practical potential to steer model behavior across practical settings. (more detailed discussion about this experiment can be found in our response to reviewer 1jwn).
>
>
> > Could chunk-based interpretability complement or enhance existing sparse autoencoder approaches or linear probe techniques? Any preliminary experiments combining them?
>
> We have not thought about combining chunking with SAE. Can the reviewer clarify what an example of a combination could be? And what can be the specific usage case of such a combination? In general, we view chunk-based interpretability as an alternative to SAEs.
>
> > Have the authors explored whether the extracted chunks remain stable or shift when the same base model is finetuned on different tasks? This could reveal insights into concept drift during finetuning.
>
> Our RNN experiment goes in this direction. What we did was to train the RNN with a training sequence that contains two randomly occurring words, ABCD and JKLMN, and fine-tune it on another sequence with the combined word ABCDLMN. We observed that the network state after receiving D, was not stable, and shifted to another location, as the network is fine tuned on the new ABCDLMN sequence. It looks like the network is not reusing its previous state of L, which leads to the subsequent predictions of M and N, but rather to create a new network state that uniquely represents L and relearns new subsequent states and their associations to M and N. We agree that the stability and efficient reuse of previous network states closely relates to concept drift and catastrophic forgetting.
>
> > The paper could better contextualize how its framework integrates with or contrasts against recent neuron-centric interpretability methods like SAE circuits or representation engineering.
>
> We propose to go beyond finding single neurons that encode for features, or finding lower-dimensional activities of neural subspaces, but rather look at raw population activities as a subject of interpretation. This is different from sparse autoencoders (SAEs) (Bricken et al. 2023), which trains an autoencoder to map neural activities of a hidden unit layer to a much larger number of intermediate hidden units while encouraging a sparse number of them to be active. In this way, the target hidden layer activity can be represented by a superposition of several individual neurons inside the SAE. However, SAE features often do not correspond clearly to human-interpretable concepts. The mapping between features and words is indirect, requiring additional steps such as VocabProj or MaxAct to identify associated tokens. This makes steering both complex and fragile. Interpreting SAE features remains an open challenge, and developing effective steering methods is still an active area of research.
>
> This approach is also different from the representation engineering approach that captures the distinct neural activity corresponding to the target concept or function, such as bias or truthfulness. Then, it uses a linear model to identify the neural activity direction that predicts the concept under question or interferes with the network behavior.
>
> With respect to intervention and steering, our method for steering and unlearning is simpler and more direct than that of SAEs. In SAEs, steering is typically achieved by clamping a feature to a specific value—either through a constant multiplier or a scaled value relative to its maximum activation (Templeton et al., 2024), which probabilistically influences the network’s output during text generation.
>
> In contrast to our grafting, which overwrites the neural activity with a given chunk, applying a direction through representation engineering requires specifying the strength of the applied latent direction. This hyperparameter is not always straightforward to determine.
>
> We will improve the contextualization in the updated version of the paper.
>
> > Beyond the included narrative schema experiments, could future work assess whether chunk activations correlate with human attention or comprehension on tasks like QA or summarization?
>
> This is a great suggestion. In future work, we could relate the neural chunk activation with human perception of the sequences, such as attention or correct comprehension. This could indeed be a testing ground to find neural correlates of successful question answering and relate that to human cognitive tendencies when doing such tasks.

---

### Note · Authors · 2025-08-14

This paper bridges cognitive science and interpretability by introducing a framework that interprets neural network computations through recurring population patterns, treated as perceptual entities. It proposes a method to extract such “chunks” from neural population activities, demonstrates that these chunks encode concepts, and shows that manipulating them can alter network behavior.

The paper was well received during the first round of review, with reviewers recognizing it as addressing a key problem in interpretability through a compelling and novel framework. Reviewer 86W2 described the contribution as novel and stimulating; reviewer 1jwn praised its focus on a central challenge and its compelling conceptual shifts; fcFE highlighted its complementary perspective to other interpretability methods; and Nmtb found the method impressive.

Some reviewers had reservations the method’s generality across datasets and the qualitative nature of the chunk steering evaluation. In the rebuttal, we addressed these points by quantifying chunk grafting efficiency, confirming chunk generality on benchmark datasets, clarifying the computational efficiency of our approach compared to existing methods, providing a more formal definition of the framework, and demonstrating that steering chunks also alters VLM outputs. Our additional evaluation suggests that grafting is more effective in earlier layers (1-10) and produces clearer chunks for more concrete word types. And additionally, LLM internal neural activity is compressible. These findings open a range of possibilities for future insights (fcFE).

The rebuttal experiments have largely addressed the reviewers’ concerns. Reviewers 1jwn (4→5), dfcf (3→4), Nmtb (3→4), and fcFE (4→5) expressed that they would be happy to see this paper accepted, and together with reviewer 86W2’s original rating of 5, the overall post-rebuttal score is __55544__.

We are deeply grateful for the reviewers’ time, constructive suggestions, and support in strengthening this contribution, and we will incorporate their feedback into the revised version of the paper.

---

### Decision · Program_Chairs · 2025-09-17

**Decision:**

Accept (poster)

**Comment:**

In this submission, the authors challenge prevailing attitudes towards the uninterpretability of neural networks by arguing that they exhibit patterns that follow regularities in the training data. The authors propose the Reflection Hypothesis - the idea that the raw neural population activity of neural networks exhibits patterns that reflect the statistical regularities of the training data. Building on this insight, the authors use "chunking", a cognitively-inspired method to extract concepts in three ways: discrete sequence chunking (DSC), population averaging (PA), unsupervised chunk discovery (UCD).

Initial reviews on this paper were mixed, but after discussion, all reviewers converged to support acceptance. Reviewers generally praised the importance of the problem and the novelty of the approach.